# Goal-Conditioned Agents that Learn Everything All at Once

Michael Matthews [1]  Matthew Jackson [1]  Michael Beukman [1]  Thomas Foster [1]  Alistair Letcher [1]
Scott Fujimoto [2]  Cédric Colas [3 4]  Jakob Foerster [1]

## Abstract

A goal-conditioned reinforcement learning agent exploring an environment will see a wealth of information throughout a trajectory, most of which is discarded when only performing on-policy updates with respect to the commanded goal. *All-goals learning*, where each transition is used for learning off-policy with respect to every goal, allows agents to extract maximal information, however it is usually computationally infeasible when done via naïve relabelling. This can be overcome by jointly outputting values and actions for every goal at once, allowing for efficient, parallel all-goals updates with a single pass through the network, in a process we call Learning Everything all at Once (LEO). We show that this approach significantly outperforms other methods on goal-conditioned Craftax and is competitive with existing baselines on continuous control environments, while achieving a $> 250\times$ speed-up compared to all-goals relabelling. We then go on to show that this approach can be made even more powerful by using LEO as a teacher network, rather than a direct actor. We hope that, by unlocking all-goals learning at scale, LEO can serve as a useful tool for RL practitioners in complex environments. We open source our code[1].

## 1. Introduction

Goal-Conditioned Reinforcement Learning (GCRL) offers a promising route to learning general and steerable policies that can execute a range of tasks upon command. Furthermore, the notion of goal conditioning lends itself well to efficient data reuse, by considering how the completion

of one goal may affect the completion of others. Imagine attempting the goal to *"walk to the supermarket"*. In attempting this task you will likely also incidentally observe information useful for completing other goals. For instance you may walk past the dentist on the way, or pass a road that you already know leads to the gym. If you were later commanded to perform a different goal, you should be able to reframe your prior experience in the context of this new goal, even if that prior experience was gathered in service of a different destination. In GCRL, this is typically done through off-policy learning on the trajectory by *relabelling* it with a different goal to that which was used to induce the trajectory. As well as using our first trajectory as a positive example for reaching the supermarket, we could also view it as a successful example for *"walk to the dentist"*, an unsuccessful but useful trajectory for *"walk to the gym"* and a negative example for *"walk to the office"*. A goal is like a lens through which to view an experience, and by viewing it through many different lenses we can multiply the information we extract.

Indeed, the larger our goal set, the more information we can theoretically extract from each gathered transition. In the case of finite goal sets, we could consider every transition with respect to every goal. While early work in GCRL recognised the value in updating with respect to all goals (Kaelbling, 1993; Sutton et al., 2011), this approach has largely fallen from favour, with most modern approaches updating with respect to a single or a small number of goals. Why are we willingly throwing away so much information?

The answer is one of computational constraints. Naïve goal relabelling scales linearly with the number of goals, meaning it is often impractical to perform more than a handful of times. Furthermore, while in theory there is information to be gained from updating with respect to every goal, in practice we see that relabelling with a single informative goal can strike a good balance between speed and performance. Hindsight Experience Replay (HER; Andrychowicz et al. (2017)), which relabels with a goal that is achieved later in the episode and thus guarantees positive signal, has become the ubiquitous method for goal relabelling in binary terminal reward settings.

[1]University of Oxford [2]McGill University [3]MIT [4]Inria. Correspondence to: Michael Matthews <michael.matthews@eng.ox.ac.uk>.

*Proceedings of the 43rd International Conference on Machine Learning*, Seoul, South Korea. PMLR 306, 2026. Copyright 2026 by the author(s).

[1]https://github.com/MichaelTMatthews/purejaxgcrl

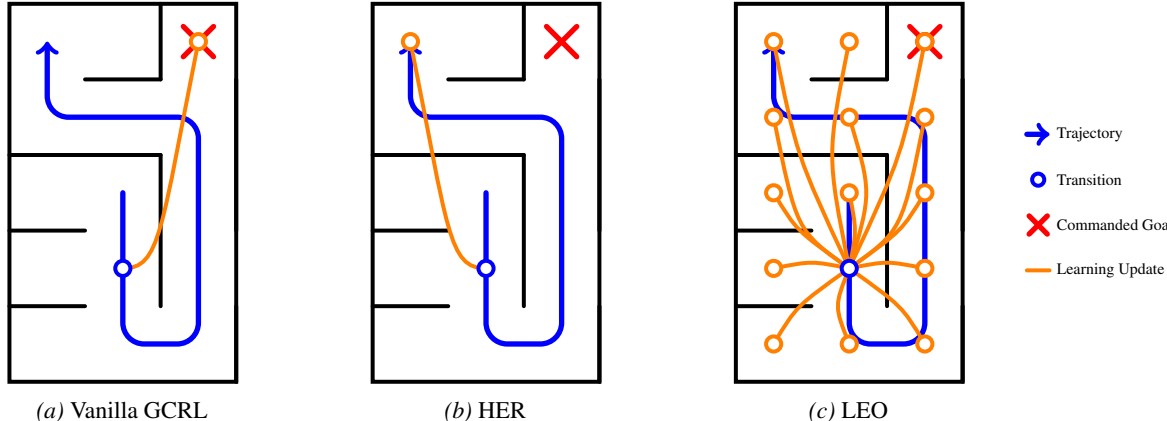

*(a) Vanilla GCRL*         *(b) HER*         *(c) LEO*

*Figure 1.* We consider with respect to what goal a given transition in a trajectory is updated to in different GCRL paradigms. In vanilla GCRL (1a), the update is done with respect to the goal that was commanded, even if this goal is never satisfied. When using HER (1b), the trajectory is relabelled with a goal that was achieved later on, providing a positive signal. With LEO (1c), we propose updating jointly with respect to the entire goal set using an efficient update for all-goals learning.

Rather than trying to find informative goals for relabelling, we approach the problem from the other direction. By massively scaling the number of goals we relabel with, we sidestep the problem of goal selection and just rely on the sheer number of goals to provide an informative signal, with the knowledge that the crucial goals will be included. Q-Map (Pardo et al., 2018) proposed jointly learning a 2D map of Q-values for navigation in pixel-based environments with all-goals updates using a convolutional network. We first expand this idea to encompass the more general paradigm of learning on an arbitrary discrete goal set. Specifically, we reparameterise our value network from one that conditions on a goal, to one that outputs with respect to every goal at once. This allows us to perform efficient, all-goals updates in the learning step and to act with respect to a commanded goal by simply indexing the output. For actor-critic algorithms, we can similarly adapt the policy network to an all-goals policy. We call this method Learn Everything all at Once (LEO).

However, while we find that LEO can indeed greatly improve performance, it can also counter-intuitively actually hurt performance on some goals. We posit that this is due to a form of late fusion (Karpathy et al., 2014), where the commanded goal is not known until the decomposition into goal heads, forcing the network to learn a representation suitable for all goals and forming an information bottleneck.

To overcome this problem, we propose combining LEO with the standard goal-conditioned approach by simply training both a LEO and goal-conditioned network independently on the same stream of data. We then consider two variants for transmitting information from the LEO teacher network to the student network: one based on policy/value cloning and one based on value interpolation. We find that this approach significantly improves on using either network

by itself. Intuitively, the LEO network performs the job of internalising maximal information at a coarse level. It provides directionally useful (but imprecise) guidance for the goal-conditioned network, which can learn more accurate estimates once it has seen some positive examples to bootstrap off. We refer to this approach as Dual LEO.

To evaluate LEO, we adapt the Craftax environment (Matthews et al., 2024a) into a challenging goal-conditioned benchmark with a large heterogeneous set of goals. In contrast to most existing GCRL benchmarks, since Craftax is both partially observed and procedurally generated, it is not meaningful to use target states as goals. We also evaluate on traditional continuous control robotics environments (Bortkiewicz et al., 2024).

In summary, our contributions are

1. We propose Learning Everything all at Once (LEO): an approach that allows us to perform efficient all-goals updates on arbitrary discrete goal sets.

2. We introduce a GCRL benchmark based on Craftax.

3. We show that LEO has a limitation caused by late fusion and propose instead using LEO as a teacher, in a method we call Dual LEO.

4. We extend and apply LEO to continuous control environments, showing the versatility of the concept.

## 2. Background

### 2.1. Goal-Conditioned Reinforcement Learning

We consider the standard reinforcement learning (RL) framework (Sutton & Barto, 1998) augmented with goal conditioning. We define a Goal-Conditioned Markov Decision Pro-

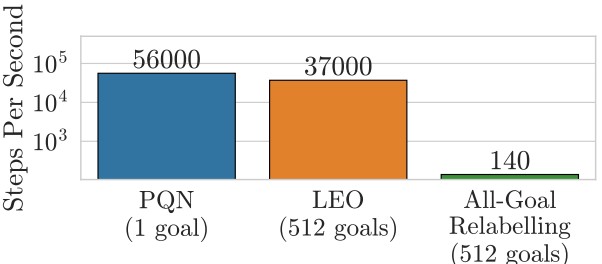

*Figure 2.* Speed comparison of different methods on the CraftaxGC benchmark with a goal set of size 512. We see that LEO learns with respect to the entire goal set with only a 34% slowdown compared to regular single goal learning. This is in contrast to naïve all-goals relabelling, which grows each batch of trajectories by a factor of 512, resulting in $264\times$ slower throughput than LEO. All methods here use PQN as their underlying algorithm with the same hyperparameters for a fair speed comparison on a single L40S GPU.

cess as $\langle \mathcal{S}, \mathcal{A}, \mathcal{R}, \mathcal{T}, \mathcal{G}, \mathcal{Z}, p_0 \rangle$, where $\mathcal{S}$ is the set of states, $\mathcal{A}$ is the set of actions, $\mathcal{T} : \mathcal{S} \times \mathcal{A} \to \Delta(\mathcal{S})$ is the stochastic transition function and $p_0 : \Delta(\mathcal{S})$ is the initial state distribution. $\mathcal{G}$ represents the set of goals, $\mathcal{R} : \mathcal{S} \times \mathcal{G} \to \mathbb{R}$ is the goal-conditioned reward function and $\mathcal{Z} : \mathcal{S} \times \mathcal{G} \to \mathbb{B}$ is the goal-conditioned pseudo-termination function (Sutton et al., 2011).

A goal-conditioned policy $\pi : \mathcal{S} \times \mathcal{G} \to \Delta(\mathcal{A})$ is trained to maximise its expected discounted return under the uniform goal distribution $\mathbb{E}_{g \sim \mathbb{U}[\mathcal{G}]}[\sum_t \gamma^t \mathcal{R}(s_t, g)]$ where $\gamma$ is the discount factor (Schaul et al., 2015). Trajectories are gathered by sampling an initial state $s_0 \sim p_0$ and goal $g \sim \mathbb{U}[\mathcal{G}]$ and then sequentially sampling actions $a \sim \pi(s_t, g)$ and states $s_{t+1} \sim \mathcal{T}(s_t, a_t)$. We refer to the notion of acting towards a specific goal $g$ as *commanding* that goal.

We follow Sutton et al. (2011) and Schaul et al. (2015) in allowing goals to be arbitrary reward functions, rather than restricting to binary terminal rewards as is sometimes done (Eysenbach et al., 2022). This setting is also sometimes referred to as multi-task RL (Andreas et al., 2017).

### 2.2. Goal Relabelling

Since the commanded goal has no effect on transition dynamics, a trajectory $(g, s_1, a_1, r_1, ..., s_T, a_T, r_T)$ gathered by commanding goal $g$ can be *relabelled* with some other goal $g'$ to $(g', s_1, a_1, \mathcal{R}(s_2, g'), ..., s_T, a_T, \mathcal{R}(s_{T+1}, g'))$, assuming that we have oracle access to the goal-conditioned reward function (Kaelbling, 1993). Since in general $\pi(s, g) \neq \pi(s, g')$, the relabelling of a trajectory makes it off-policy.

Various methods for goal relabelling have been proposed, including by random uniform sampling of goals (Schaul et al., 2015) and through sampling from a generative model (Nair

et al., 2018). The most common form of goal relabelling is Hindsight Experience Replay (HER; Andrychowicz et al. (2017)). HER makes the assumptions that (1) each goal is associated with a binary reward and (2) we have access to some function $f : \mathcal{S} \to \mathcal{G}$ that maps states to the unique goal that is achieved by a state. With these assumptions, we can obtain a positive signal from trajectories that failed to reach their target, for instance by relabelling with the final state $f(s_T)$. We later investigate settings where these assumptions do not hold (Section 4).

### 2.3. All-Goals Updates

All-goals updates for RL were first proposed in tabular settings by Kaelbling (1993), where a discrete set of value functions would be updated online from every transition. This idea was combined with function approximation in Horde (Sutton et al., 2011), where multiple 'demons' learn from every update with a separate linear projection from a hardcoded embedding. Universal Value Function Approximators (UVFA; Schaul et al. (2015)) proposed learning an 'infinite horde' that conditions on a representation of the goal function and updates by off-policy learning with many (not all) sampled goals. Q-Map (Pardo et al., 2018) later proposed applying an all-goals update with a convolutional network on pixel arrays to learn reachability to each pixel. The basic LEO method could be considered a generalisation of Q-Map to arbitrary discrete goal sets or equivalently as a Horde network with a learned shared embedding and parallelised updates.

## 3. Learning Everything All at Once

In this section we first introduce the LEO method in the discrete action setting (Section 3.1), before discussing some disadvantages of LEO and proposing Dual LEO as a method to overcome these (Section 3.2). Finally we adapt LEO for continuous control settings (Section 3.3).

### 3.1. Efficient All-Goals Updates

We first consider the problem of learning a goal-conditioned Q-network over a finite goal set $\mathcal{G}$ in the discrete action case. The traditional approach would be to learn a function that conditions on both the state and the goal and outputs an array of Q-values, one for each discrete action $Q(s, g) : \mathcal{S} \times \mathcal{G} \to \mathbb{R}^{\mathcal{A}}$. We refer to this method as UVFA (Schaul et al., 2015) or UVFA-style. Given a tuple $(s, a, s')$ we could then perform the off-policy Q-learning update (Watkins & Dayan, 1992; Mnih et al., 2013) with loss defined as $\mathcal{L}(g) = (\mathcal{R}(s', g) + \gamma \cdot \max_{a'} Q(a'|s', g) - Q(a|s, g))^2$ for some goal $g$ that is not necessarily the commanded goal.

Assuming oracle access to the goal-conditioned reward function, we could relabel the trajectory as many times as we'd

like, potentially multiplying the information content, without requiring any extra samples from the environment. Since we have assumed a finite goal set, we can take this to its extreme and update on all experience with every goal.

However, simply relabelling each transition with every goal quickly becomes intractable for non-trivial goal sets, as the computational costs of relabelling scale linearly. To overcome this, we instead propose learning an all-goals Q-function $Q(s) : \mathcal{S} \to \mathbb{R}^{\mathcal{G} \times \mathcal{A}}$. Rather than outputting a 1D array of Q-values (over actions) for each input state and goal, the all-goals Q-function outputs a 2D array of Q-values (over actions and goals) for each input state. This process is known as *currying*, which transforms any function $f : X \times Y \to Z$ to a function $\mathrm{curry}(f) : X \to Z^Y$, where $Z^Y$ denotes functions $Y \to Z$.

We can then vectorise the Q-learning update to arrive at the all-goals update defined by minimising the loss:

$$\mathcal{L} = (\boldsymbol{\mathcal{R}}(s') + \gamma \cdot \max_{a'} \boldsymbol{Q}(a'|s') - \boldsymbol{Q}(a|s))^2 \,,$$

where $\boldsymbol{\mathcal{R}}(s')$ is the vector of rewards across goals upon entering state $s'$. Note that the update looks like a regular (i.e. not goal-conditioned) Q-learning update and makes no reference to the goal that was commanded or some notion of a relabelled goal, but operates purely on $(s, a, s')$ tuples.

We call this method Learn Everything all at Once (LEO).

### 3.2. Dual LEO

While the LEO architecture allows for efficient all-goals updates, this comes with a tradeoff to representational capacity. Consider using a LEO Q-network at inference time to take greedy actions. While we may know a priori which goal we wish to command, the network does not, and therefore must produce outputs for every goal and assign equal weighting of its finite computation to each one, even though we will throw away the outputs for all but one goal. What is a strength of the network while learning is a hindrance when acting. This can be seen as a form of *late fusion* (Karpathy et al., 2014) of the state and goal modalities and stands in contrast to the early fusion of UVFA networks, where the goal is fed into the network from the beginning.

To overcome this, we propose training both a LEO and UVFA network simultaneously. The LEO network can act as a "data sponge" that does not miss any crucial transitions with respect to any goal, while the UVFA network learns high fidelity estimates for the *commanded* goal. We call this approach Dual LEO. We consider two variants:

**Dual LEO (PQN)**
We train a LEO Q-network and a UVFA Q-network off-policy on the same stream of data. Q-value estimates for acting are then taken as the linear combination of two net-

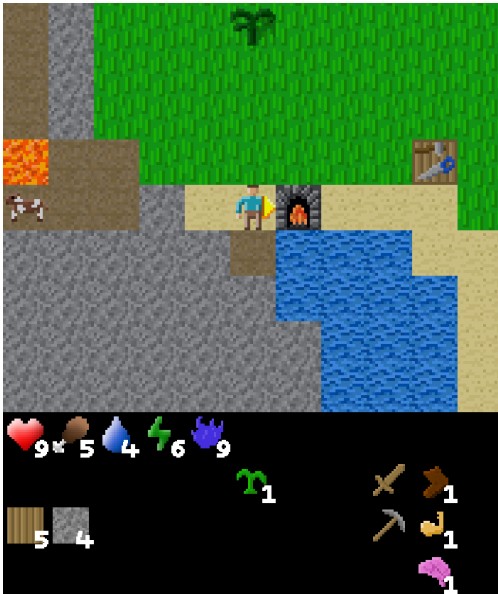

*Figure 3.* Example frame from CraftaxGC. Goals completed in this frame include `tools/stone_pickaxe`, `inventory/wood_5` and `block_map/furnace_right`.

works $\alpha \cdot Q_{\mathrm{LEO}}(s, a, g) + (1 - \alpha) \cdot Q_{\mathrm{UVFA}}(s, a, g)$ where $\alpha$ is the mixing coefficient. Each network is updated independently, using its own estimates for bootstrapping.

Consider a hard goal for which we have never seen a commanded solution to, that is incidentally achieved while *not* being commanded. The LEO network will update with this information and form a rough value estimate that, while not perfect, may be enough to sometimes achieve the goal. Since a commanded solution has never been seen, the UVFA network should have all its Q-estimates close to zero. When this goal is eventually sampled and commanded, the linear combination will therefore be proportional to just the LEO Q-estimates, allowing for the goal to be achieved. This provides a commanded example of the goal being solved, giving signal to the UVFA network to start forming its own value estimates. If $\alpha$ is small, as the UVFA network learns and the magnitude of its Q-estimates increase, the linear combination will come close to just the high fidelity UVFA approximation. We later empirically validate that this sequence of events occurs in Figure 6 in Section 6.2.

In this way, LEO acts as a teacher for the UVFA network for goals that the UVFA network cannot solve yet, before largely surrendering control to the UVFA network once it has learned to solve them (assuming a small $\alpha$).

**Dual LEO (PPO)**
A PPO actor-critic network is trained as per normal, with actions taking by the PPO actor, while a LEO network trains off-policy on the same generated stream of data.

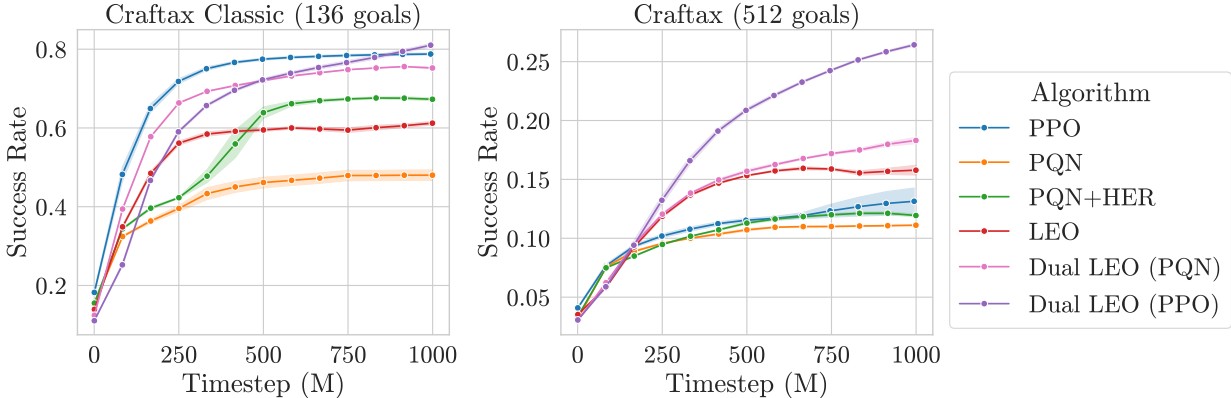

*Figure 4.* Mean success rate across all goals on CraftaxGC. The shaded area denotes 1 standard error over 5 seeds. We see LEO outperforming UVFA-style baselines on the larger goal set, but not on the smaller one. Dual LEO performs well in both cases, with the PPO variant achieving the best final performance in both settings.

We then add losses to push the PPO policy $\pi(s, g)$ towards the greedy LEO policy $\operatorname{argmax}_a Q_{\text{LEO}}(s, a, g)$ and the value network $V(s, g)$ towards the LEO value estimate $\max_a Q_{\text{LEO}}(s, a, g)$.

Similar to the PQN variant, the LEO network will form better estimates for goals that have only been seen uncommanded, and pass on this information to the UVFA-style PPO actor-critic network through the additional loss term.

### 3.3. LEO for Continuous Control

LEO can naturally be extended to actor-critic algorithms by similarly currying the goals from the observation space to the output space in the policy network to obtain an all-goals policy $\pi : \mathcal{S} \to \Delta(\mathcal{A})^{\mathcal{G}}$.

First, we can define the all-goals critic update by swapping the maximisation in favour of actions selected by the all-goals policy:

$$\mathcal{L}_Q = (\mathcal{R}(s') + \gamma \cdot \boldsymbol{Q}(s', \boldsymbol{\pi}(s')) - \boldsymbol{Q}(s, a))^2,$$

where the all-goals critic update can be performed with a single backwards pass through the Q-network. The DPG (Silver et al., 2014) policy update can similarly be defined:

$$\mathcal{L}_\pi = -\boldsymbol{Q}(s, \boldsymbol{\pi}(s)).$$

However, since each goal will generally induce a different action distribution from the policy network, the policy update cannot be done with a single backwards pass, forcing us to do $\mathcal{O}(\mathcal{G})$ backwards passes through the Q-network. While one possible correction to this concern is through the use of importance sampling and alternate policy gradient updates (Nair et al., 2020), we rely on the simplest approach as a proof-of-concept of our method to continuous actions.

## 4. CraftaxGC

Most existing goal-conditioned benchmarks occupy a constrained corner of the design space: goals are typically target states or observations, with a functional (many to one) mapping from states to goals (Nair et al., 2018; Plappert et al., 2018; Fu et al., 2020; Bortkiewicz et al., 2024; Park et al., 2024). An archetypal example is Ant Maze (Fu et al., 2020), where goals are the $(x, y)$ coordinates of the ant body.

In practice, we often want goals defined at higher levels of abstraction — *"wash up the dishes"* or *"fold the laundry"* — where many different states could satisfy the same goal, and a single state might satisfy multiple goals simultaneously. Methods that assume a functional state-goal mapping, such as Contrastive RL, cannot natively handle this typical and desirable setting (Eysenbach et al., 2022).

Craftax (Matthews et al., 2024a), a partially observed, procedurally generated environment inspired by Crafter (Hafner, 2021) and the NetHack Learning Environment (Küttler et al., 2020), offers a natural testbed for this more general formulation. Because worlds are freshly generated each episode, state-based goals are meaningless: the agent will never see the same observation twice. We adapt Craftax into a goal-conditioned benchmark by defining a large set of semantic conditions: having different amounts of each item in inventory, being adjacent to specific blocks, items, or creatures, obtaining tools and weapons, reaching dungeon levels, and acquiring experience points or enchantments; see Figure 3. Each goal can be verified directly from an observation without access to latent variables.

This leads to a large, heterogeneous set of goals that can be short-horizon (*"stand next to a tree"*) or long-horizon (*"stand next to the end-game boss"*), easy (*"collect 1 wood"*), hard (*"craft a diamond pickaxe"*), and induce the

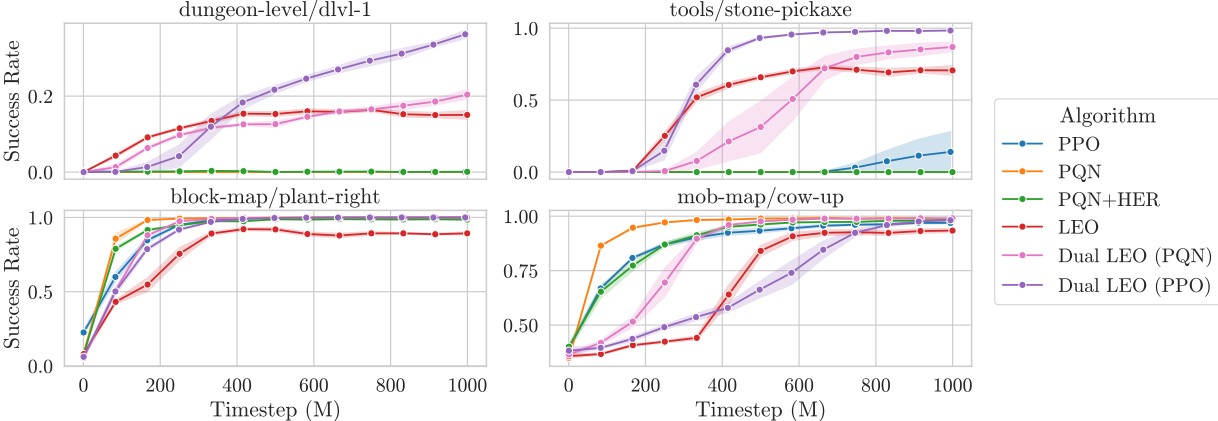

*Figure 5.* Mean success rate over selected goals on CraftaxGC. The shaded area denotes 1 standard error over 5 seeds. LEO performs well on hard goals (top row) but can underperform on easy goals (bottom row), due to the late fusion issue. Dual LEO resolves this problem, achieving strong results on the hard goals without sacrificing performance on easy goals.

agent to take one path in the game (*"grow a plant"*) or a wildly different one (*"reach dungeon level 5"*). We adapt both the full game of Craftax and the much simpler Craftax-Classic for goal-conditioned learning, with the two settings having goal spaces of size 512 and 136 respectively. For a complete listing of goals see Appendix A and Appendix B.

While CraftaxGC serves as a challenging testbed for evaluating GCRL algorithms, it can also be seen as a fundamentally different way of solving the underlying Craftax benchmark. The original environment has a small sparse set of achievements, which when all completed will naturally lead an agent towards winning the game. However, as shown by the failure of any existing method to solve the benchmark, these achievements are potentially too sparse or poor signal. The goals in CraftaxGC are a significantly larger set than the achievements and differ in that many of them are mutually exclusive (e.g. you cannot have both 4 and 5 wood at once). By training an agent able to achieve any goal *upon command*, rather than simply maximising the sum of goals/achievements, we provide an alternative route to solving the benchmark that naturally elicits exploration and state coverage. Any agent that solves CraftaxGC could in theory essentially solve the underlying Craftax environment by commanding the goal `dungeon_level/dlvl_8`.

## 5. Experimental Setup

We evaluate on both the proposed CraftaxGC environments, as well as the ant maze tasks from JaxGCRL (Bortkiewicz et al., 2024).

### 5.1. CraftaxGC

We build LEO off of PQN (Gallici et al., 2024), an off-policy algorithm with strong results on the original Craftax benchmark. As baselines we consider PQN with and without HER, as well as PPO (Schulman et al., 2017). We then combine these approaches in the PQN and PPO variants of Dual LEO (note that the LEO component is based off PQN in both cases). Hyperparameters for all methods were tuned with equal budgets (Appendix D). We also swept over HER strategies (Appendix E) and report the best.

Since CraftaxGC episodic returns lie in $[0, 1]$, we bound all value estimates with a sigmoid. PQN relies on layer normalisation rather than target networks to control value overestimation, which works well enough for near-on-policy data but not for LEO's highly off-policy updates. Without bounded estimates, LEO's Q-values frequently diverged. We apply this change to all PQN-based methods and to PPO's value network for fairness.

We further use a simple autocurriculum: commanded goals are sampled only from goals observed at least once in the past, to prevent completely out of reach goals being sampled. Without this curriculum, LEO and Dual LEO actually even further outperform the baselines on Craftax (see Appendix F), but we include the curriculum in our main results as this would be such a simple and obvious modification for any practitioner trying to achieve strong results.

Goals are sampled uniformly (from the set of previously seen goals) at the beginning of each episode. When a goal is achieved we follow the "first return then explore" paradigm (Ecoffet et al., 2021) by sampling a new goal and inserting a pseudo-termination flag, which is equivalent to starting a new episode in the given state.

We did also create a variant of HER that relabels with all goals, however we found it far too slow to be practical (Figure 2), so do not include any results for this method.

## 5.2. JaxGCRL Continuous Control

We adapted LEO for continuous control and applied it to the ant maze tasks from the JaxGCRL benchmark (Bortkiewicz et al., 2024). The natural goal sets for these tasks are the infinite set of $\mathbb{R}^2$ vectors that lie inside the environment boundaries. This would seemingly prohibit LEO from being applicable: we cannot have a network with infinite heads. However, reaching any of these goals exactly is generally impossible, which is why JaxGCRL and similar environments evaluate a successful trajectory as one that comes within some predefined $\epsilon$ of the goal location. Bearing this in mind, we can place a discrete grid of goals on the plane and snap any continuous goal to the closest quantised goal on the grid. If the grid has a high enough fidelity, then reaching the closest quantised goal will guarantee also solving the true continuous goal, allowing us to effectively employ LEO for these tasks. If exact goal reaching is important, the error between the real and quantised goal could be fed in as an observation, providing a form of goal reaching that is somewhere between UVFA and LEO, although we do not investigate this approach.

We adapt the Soft Actor-Critic algorithm (SAC; Haarnoja et al. (2018)) by replacing both the Q-networks and policy network with LEO networks, as described in Sections 3.1 and 3.3. We compare against goal-conditioned SAC and TD3 (Fujimoto et al., 2018) using standard UVFA architectures, with and without HER, as well as Contrastive RL (Eysenbach et al., 2022). All methods use the 'small' architecture from Bortkiewicz et al. (2024) (2 layers, width 256) with default hyperparameters. We perform no additional tuning for LEO.

## 6. Experimental Results

### 6.1. CraftaxGC

**Craftax-Classic** The results for mean success rate over all goals are shown in Figure 4. We see that for the small goal set of Craftax-Classic, a simple goal-conditioned PPO agent that updates only on commanded goals performs relatively well. We see that PQN performs significantly worse than PPO, but improves with the addition of HER. LEO shows stronger performance than its base algorithm PQN, but is weaker than PQN+HER. Both variants of Dual LEO perform better than their constituent parts, with the PPO variant marginally becoming the best performing method. Overall, the results on Craftax-Classic show that simple algorithms may be enough when dealing with small sets of relatively simple goals and we include them to provide a fair picture.

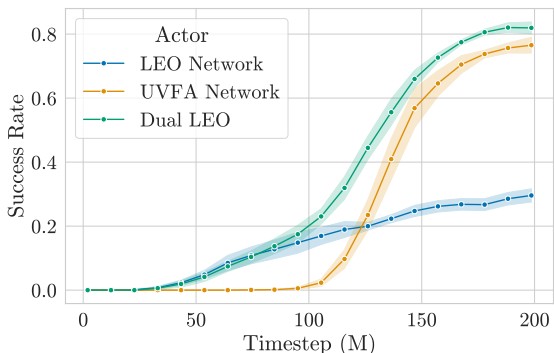

*Figure 6.* Mean success rate for the `inventory/coal-1` goal for Dual LEO (PQN), when acting greedily with respect to each of its components. The shaded area denotes 1 standard error over 5 seeds. Validating our hypothesis in Section 3.2, we see that the LEO network learns to achieve the goal early, providing positive examples of goal completion that allows the UVFA network to learn on.

**Craftax** On the full Craftax benchmark the results look very different. PQN and PPO, which update only with commanded goals, both perform quite poorly, with HER marginally improving the PQN results. LEO gives a significant boost, with Dual LEO (PQN) performing even better, and Dual LEO (PPO) massively outperforming all other methods. Taking a closer look at selected individual goals, we see a more nuanced picture (Figure 5). While LEO learns to make progress on many hard goals, it often performs worse than the baselines on easier goals. UVFA-style methods that include the commanded goal as an observation can use the whole forward pass to focus on that single goal, whereas LEO must spread its computation over the entire goal set, likely resulting in these discrepancies. We see that Dual LEO overcomes this issue by combining the best of LEO-style and UVFA-style methods by doing well on the hard goals, without sacrificing performance on easy goals.

Note that, even with the goal sampling being reduced to previously seen goals, the majority of trajectories do not reach their commanded goal. PQN and PPO can learn very little from such episodes as they see no positive signal (that is not to say that negative signal is worthless — it just tends to be much more common than positive goal completion signal). In contrast, LEO can make full use of these episodes, as its updates are agnostic to the commanded goal. We see LEO converge around 500M timesteps, whereas Dual LEO continues to improve, showing that the LEO network has given enough positive signal for the UVFA network to warm start off, while LEO may have saturated its bottleneck.

We take a more precise look at how the different algorithms respond to a varying goal set size in Appendix G.

## 6.2. LEO as a Teacher

We now take a closer look at the nature of Dual LEO (specifically the PQN variant), by considering how each of its components performs when acting greedily (Figure 6), where we see the teacher-student dynamic that was proposed in Section 3.2 in action. We use the `inventory/coal-1` goal as an example of a goal that is solved by Dual LEO (PQN), but not by PQN (see Appendix C), indicating that the LEO network is helping in some way. In Figure 6 we see the LEO component learns to solve the `inventory/coal-1` goal early but suboptimally. However this seems to be enough to allow the Dual LEO actor to gather positive examples, which then facilitates the UVFA component learning a strong policy.

The bad final performance of the LEO component, combined with the strong UVFA performance (and the fact that UVFA by itself does not ever achieve this goal), provides a strong positive signal for using LEO as a teacher.

Further analysis is in Appendix H.

## 6.3. JaxGCRL

Figure 7 shows that SAC+LEO outperforms all baselines on the smaller U Maze. On the larger maze the results are less clear, with LEO, SAC+HER and CRL all performing similarly. We also investigated using a Dual LEO critic for SAC, but found it did not noticeably affect performance. This could be because the main difficulty in the ant maze tasks is learning the locomotive gait, rather than the differing goal positions. This is in contrast to Craftax where the different goals initiate very different and often opposing behaviours, with the low level locomotive task largely abstracted away.

Furthermore, it should be noted that we use a sparse reward for goal reaching, as done in the original JaxGCRL paper. However, an advantage of LEO over approaches like CRL is that it is suitable for arbitrary reward functions, that could include a dense distance reward and shaping terms. However, for a fair alignment with the original benchmark proposal, we do not explore this.

## 7. Related Work

### Goal-Conditioned RL

GCRL (Kaelbling, 1993) is the augmentation of the RL paradigm with the ability to command goals to elicit different behaviours, rather than having the RL agent optimise a single reward function. GCRL has been studied largely in robotic settings (Andrychowicz et al., 2017; Ghosh et al., 2019; Eysenbach et al., 2020) for both online and offline (Ghosh et al., 2019; Peng et al., 2019; Lynch et al., 2020; Park et al., 2023; 2024) RL. Related is Hierarchical RL, which can be seen as learning both a

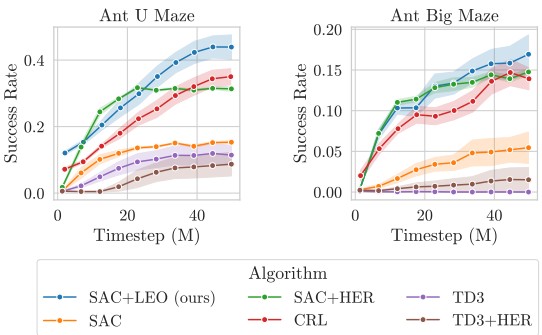

*Figure 7.* Mean success rate on JaxGCRL ant maze tasks for 50 million timesteps. The shaded area denotes 1 standard error over 5 seeds. Note that although LEO quantises the goal space internally, goal sampling for training and evaluation is done on the true underlying continuous goals just as with the other methods.

goal-conditioned agent and a high level agent to sequence goals (Dayan & Hinton, 1992; Sutton et al., 1999; Precup, 2000; Bacon et al., 2017; Vezhnevets et al., 2017; Klissarov & Precup, 2021; Chen et al., 2023; Klissarov et al., 2024; Park et al., 2025; Henaff et al., 2025; Klissarov et al., 2025). Successor features (Dayan, 1993; Barreto et al., 2017) and specifically universal successor features (Borsa et al., 2018) can be considered a more general paradigm than what we study, where our discrete GCRL setting limits the weight vector to being one-hot.

### Many-goals RL updates

While all-goals updates have often been seen as expensive or infeasible, prior work has made use of many-goal updates. UVFA (Schaul et al., 2015) proposed using each transition to update with as many randomly sampled goals as your budget would allow. Many-goals RL (Veeriah et al., 2018) proposed updating efficiently with multiple goals by learning separate state and goal embeddings. Contrastive RL (Eysenbach et al., 2022) performs an efficient many-goals update through the contrastive learning step, where each positive future sample is contrasted with many negative random samples. We investigate using the LEO framework for many-goals updates in Appendix I.

### Exploration in RL

As well as a route to general agents, GCRL can be seen as natively encouraging exploration through commanding a diverse set of goals. Inducing novel behaviour has been explored in both goal-conditioned (Florensa et al., 2018; Colas et al., 2018; Pong et al., 2019; Blaes et al., 2019; Zhang et al., 2020; Ecoffet et al., 2021; Colas et al., 2022) and single task (Bellemare et al., 2016; Achiam & Sastry, 2017; Pathak et al., 2017; Burda et al., 2018; Pathak et al., 2019) settings.

**Similar Architectures**

The idea of using goal-conditioned auxiliary tasks (Jaderberg et al., 2016; Mirowski et al., 2016; Fedus et al., 2019) arrives at a similar framework to both LEO and Q-Map (Pardo et al., 2018), where Q-estimates for multiple goals are learned in parallel. However, the purpose of these goals is not actually for acting on but to provide a self-supervised signal for representation learning, in service of a single target policy.

Another similar architecture is MT-MH-SAC (Yu et al., 2020; Yang et al., 2020), where each task is assigned its own head in a multi-task setting. In contrast to LEO, each head is only updated with respect to its own task, and the task representation is fed in as input to the network.

**Environments for GCRL**

The notion of what constitutes a specifically *goal-conditioned* RL environment is not entirely clear. If you do not implement techniques that make use of the separability of the goal from the observation (e.g. LEO, HER, Contrastive RL), a goal-conditioned environment is indistinguishable from a regular RL environment. Environments that often specifically make use of these techniques for robotics include Fetch (Plappert et al., 2018), Ant Maze (Fu et al., 2020), JaxGCRL (Bortkiewicz et al., 2024), OG-Bench (Park et al., 2024) and BuilderBench (Ghugare et al., 2025). Minecraft has been used for goal-conditioned RL in various forms (Guss et al., 2019; Fan et al., 2022; Lifshitz et al., 2023; Wang et al., 2023). Other environments that are goal-conditioned *in spirit*, i.e. you could meaningfully separate out a goal include Minigrid (Chevalier-Boisvert et al., 2023), Kinetix (Matthews et al., 2024b) and XLand-minigrid (Nikulin et al., 2024).

**Goal Modalities**

The most common goal modality in prior work is to use the observation or a fixed subset of the observation as the goal representation (Nair et al., 2018; Plappert et al., 2018; Fu et al., 2020; Bortkiewicz et al., 2024; Park et al., 2024). Since the advent of Vision-Language-Action Models (Zitkovich et al., 2023), natural language has emerged as an increasingly powerful goal modality in robotics (Black et al., 2024; Kim et al., 2024; Team et al., 2025; Intelligence et al., 2025), but also applied to other domains like video games (Klissarov et al., 2023; Wang et al., 2023; Klissarov et al., 2024). There is also work looking at using goals specified in temporal logic (Vaezipoor et al., 2021; Icarte et al., 2022; Yalcinkaya et al., 2024) and learned latent spaces (Touati & Ollivier, 2021; Gallouédec & Dellandréa, 2023; Tirinzoni et al., 2025).

## 8. Limitations and Future Work

While we have shown that LEO can provide compelling empirical results in suitable settings, the method does have clear limitations. Firstly, it assumes a finite goal set that is small enough that it can be curried to the output layer of value/policy networks, making it unsuitable for very large or continuous goal spaces. Also, LEO represents goals as a set, which may be suboptimal if the goals have some underlying structure, for instance representing points on a grid.

While we have shown that low dimensional continuous goal spaces can be effectively quantised into a discrete goal set, this approach would not scale to high-dimensional continuous goal spaces. Furthermore, the need to resample actions for each goal head in the actor-critic setup for the policy update is a significant slowdown for continuous control settings (we found the all-goals policy update to drop throughput by about 70%). This could be improved upon by making use of algorithms that reuse actions from the gathered trajectory batch (Nair et al., 2020) rather than resampling.

We hope that we have shed light onto what we believe is a core tradeoff in GCRL, between 1) the UVFA-style approach which can generalise between goals and learn good representations through early fusion and 2) the LEO-style approach, which can efficiently ingest a huge amount of information, but must treat each goal independently and can struggle to learn high fidelity representations due to late fusion. Dual LEO attempts to bridge this gap by combining the two paradigms, but perhaps some method exists that could natively interpolate between the two extremes.

Future work could investigate integrating LEO with other frameworks, such as those from hierarchical RL, dynamically constructing and adding goals to the LEO network throughout training or using the Dual LEO setup as a signal for goal sampling.

## 9. Conclusion

In conclusion, we propose LEO, an approach that allows efficient all-goals updates for finite goal sets. We demonstrate that LEO outperforms existing approaches to GCRL on the full CraftaxGC benchmark, with Dual LEO, an approach that combines the best of LEO and UVFA, performing even better. Furthermore, we show that the LEO concept can be extended to continuous control, where it is competitive with existing baselines on goal-conditioned ant maze tasks. We hope that we have demonstrated the utility of LEO and that it proves to be a useful tool for RL practitioners.

## Impact Statement

This paper presents work whose goal is to advance the field of machine learning. There are many potential societal consequences of our work, none of which we feel must be specifically highlighted here.

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

# A. Full Craftax-Classic Goal Listing

We provide a full listing of all goals in the Craftax-Classic goal set in Tables 1 and 2. To give a rough empirical indication of goal difficulty we show the success rate of Dual LEO (PQN) after 1 billion timesteps.

| ID | Name | Success Rate | ID | Name | Success Rate |
|---|---|---|---|---|---|
| 0 | inventory/wood_1 | 1.00 | 46 | inventory/sapling_2 | 1.00 |
| 1 | inventory/wood_2 | 1.00 | 47 | inventory/sapling_3 | 1.00 |
| 2 | inventory/wood_3 | 1.00 | 48 | inventory/sapling_4 | 1.00 |
| 3 | inventory/wood_4 | 1.00 | 49 | inventory/sapling_5 | 1.00 |
| 4 | inventory/wood_5 | 0.99 | 50 | inventory/sapling_6 | 0.99 |
| 5 | inventory/wood_6 | 0.99 | 51 | inventory/sapling_7 | 0.99 |
| 6 | inventory/wood_7 | 0.99 | 52 | inventory/sapling_8 | 0.99 |
| 7 | inventory/wood_8 | 0.98 | 53 | inventory/sapling_9 | 0.98 |
| 8 | inventory/wood_9 | 0.98 | 54 | tools/wood_pickaxe | 1.00 |
| 9 | inventory/stone_1 | 1.00 | 55 | tools/stone_pickaxe | 0.91 |
| 10 | inventory/stone_2 | 1.00 | 56 | tools/iron_pickaxe | 0.00 |
| 11 | inventory/stone_3 | 1.00 | 57 | tools/wood_sword | 0.98 |
| 12 | inventory/stone_4 | 1.00 | 58 | tools/stone_sword | 0.92 |
| 13 | inventory/stone_5 | 1.00 | 59 | tools/iron_sword | 0.00 |
| 14 | inventory/stone_6 | 1.00 | 60 | block_map/OUT_OF_BOUNDS_left | 0.78 |
| 15 | inventory/stone_7 | 0.99 | 61 | block_map/OUT_OF_BOUNDS_right | 0.82 |
| 16 | inventory/stone_8 | 0.99 | 62 | block_map/OUT_OF_BOUNDS_up | 0.80 |
| 17 | inventory/stone_9 | 0.99 | 63 | block_map/OUT_OF_BOUNDS_down | 0.77 |
| 18 | inventory/coal_1 | 0.98 | 64 | block_map/GRASS_left | 1.00 |
| 19 | inventory/coal_2 | 0.93 | 65 | block_map/GRASS_right | 1.00 |
| 20 | inventory/coal_3 | 0.86 | 66 | block_map/GRASS_up | 1.00 |
| 21 | inventory/coal_4 | 0.76 | 67 | block_map/GRASS_down | 1.00 |
| 22 | inventory/coal_5 | 0.41 | 68 | block_map/WATER_left | 0.96 |
| 23 | inventory/coal_6 | 0.00 | 69 | block_map/WATER_right | 0.96 |
| 24 | inventory/coal_7 | 0.00 | 70 | block_map/WATER_up | 0.96 |
| 25 | inventory/coal_8 | 0.00 | 71 | block_map/WATER_down | 0.96 |
| 26 | inventory/coal_9 | 0.00 | 72 | block_map/STONE_left | 1.00 |
| 27 | inventory/iron_1 | 0.00 | 73 | block_map/STONE_right | 1.00 |
| 28 | inventory/iron_2 | 0.00 | 74 | block_map/STONE_up | 1.00 |
| 29 | inventory/iron_3 | 0.00 | 75 | block_map/STONE_down | 1.00 |
| 30 | inventory/iron_4 | 0.00 | 76 | block_map/TREE_left | 1.00 |
| 31 | inventory/iron_5 | 0.00 | 77 | block_map/TREE_right | 1.00 |
| 32 | inventory/iron_6 | 0.00 | 78 | block_map/TREE_up | 1.00 |
| 33 | inventory/iron_7 | 0.00 | 79 | block_map/TREE_down | 1.00 |
| 34 | inventory/iron_8 | 0.00 | 80 | block_map/PATH_left | 1.00 |
| 35 | inventory/iron_9 | 0.00 | 81 | block_map/PATH_right | 1.00 |
| 36 | inventory/diamond_1 | 0.00 | 82 | block_map/PATH_up | 1.00 |
| 37 | inventory/diamond_2 | 0.00 | 83 | block_map/PATH_down | 1.00 |
| 38 | inventory/diamond_3 | 0.00 | 84 | block_map/COAL_left | 0.97 |
| 39 | inventory/diamond_4 | 0.00 | 85 | block_map/COAL_right | 0.96 |
| 40 | inventory/diamond_5 | 0.00 | 86 | block_map/COAL_up | 0.97 |
| 41 | inventory/diamond_6 | 0.00 | 87 | block_map/COAL_down | 0.97 |
| 42 | inventory/diamond_7 | 0.00 | 88 | block_map/IRON_left | 0.95 |
| 43 | inventory/diamond_8 | 0.00 | 89 | block_map/IRON_right | 0.95 |
| 44 | inventory/diamond_9 | 0.00 | 90 | block_map/IRON_up | 0.96 |
| 45 | inventory/sapling_1 | 1.00 | 91 | block_map/IRON_down | 0.95 |

*Table 1.* Craftax-Classic goal listing. Success rate shows Dual LEO (PQN) performance after 1 billion timesteps. Part 1.

| ID | Name | Success Rate |
|---|---|---|
| 92 | block_map/DIAMOND_left | 0.34 |
| 93 | block_map/DIAMOND_right | 0.36 |
| 94 | block_map/DIAMOND_up | 0.35 |
| 95 | block_map/DIAMOND_down | 0.35 |
| 96 | block_map/CRAFTING_TABLE_left | 1.00 |
| 97 | block_map/CRAFTING_TABLE_right | 1.00 |
| 98 | block_map/CRAFTING_TABLE_up | 1.00 |
| 99 | block_map/CRAFTING_TABLE_down | 1.00 |
| 100 | block_map/FURNACE_left | 1.00 |
| 101 | block_map/FURNACE_right | 0.99 |
| 102 | block_map/FURNACE_up | 1.00 |
| 103 | block_map/FURNACE_down | 1.00 |
| 104 | block_map/SAND_left | 1.00 |
| 105 | block_map/SAND_right | 1.00 |
| 106 | block_map/SAND_up | 1.00 |
| 107 | block_map/SAND_down | 1.00 |
| 108 | block_map/LAVA_left | 0.85 |
| 109 | block_map/LAVA_right | 0.85 |
| 110 | block_map/LAVA_up | 0.85 |
| 111 | block_map/LAVA_down | 0.85 |
| 112 | block_map/PLANT_left | 1.00 |
| 113 | block_map/PLANT_right | 1.00 |
| 114 | block_map/PLANT_up | 1.00 |
| 115 | block_map/PLANT_down | 1.00 |
| 116 | block_map/RIPE_PLANT_left | 0.00 |
| 117 | block_map/RIPE_PLANT_right | 0.00 |
| 118 | block_map/RIPE_PLANT_up | 0.00 |
| 119 | block_map/RIPE_PLANT_down | 0.00 |
| 120 | mob_map/zombie_left | 0.99 |
| 121 | mob_map/zombie_right | 0.99 |
| 122 | mob_map/zombie_up | 0.99 |
| 123 | mob_map/zombie_down | 0.99 |
| 124 | mob_map/cow_left | 1.00 |
| 125 | mob_map/cow_right | 1.00 |
| 126 | mob_map/cow_up | 1.00 |
| 127 | mob_map/cow_down | 1.00 |
| 128 | mob_map/skeleton_left | 0.94 |
| 129 | mob_map/skeleton_right | 0.94 |
| 130 | mob_map/skeleton_up | 0.94 |
| 131 | mob_map/skeleton_down | 0.94 |
| 132 | mob_map/arrow_left | 0.93 |
| 133 | mob_map/arrow_right | 0.93 |
| 134 | mob_map/arrow_up | 0.94 |
| 135 | mob_map/arrow_down | 0.94 |

*Table 2.* Craftax-Classic goal listing. To give an indication of goal difficulty, we show the success rate of Dual LEO (PQN) after 1 billion timesteps, averaged over 5 seeds. Part 2.

# B. Full Craftax Goal Listing

We provide a full listing of all goals in the Craftax goal set in Tables 3, 4, 5, 6, 7 and 8. To give a rough empirical indication of goal difficulty we show the success rate of Dual LEO (PQN) after 1 billion timesteps.

| ID | Name | Success Rate | ID | Name | Success Rate |
|---|---|---|---|---|---|
| 0 | inventory/wood_1 | 1.00 | 45 | inventory/sapphire_1 | 0.01 |
| 1 | inventory/wood_2 | 0.99 | 46 | inventory/sapphire_2 | 0.00 |
| 2 | inventory/wood_3 | 0.99 | 47 | inventory/sapphire_3 | 0.00 |
| 3 | inventory/wood_4 | 0.96 | 48 | inventory/sapphire_4 | 0.00 |
| 4 | inventory/wood_5 | 0.96 | 49 | inventory/sapphire_5 | 0.00 |
| 5 | inventory/wood_6 | 0.94 | 50 | inventory/sapphire_6 | 0.00 |
| 6 | inventory/wood_7 | 0.93 | 51 | inventory/sapphire_7 | 0.00 |
| 7 | inventory/wood_8 | 0.91 | 52 | inventory/sapphire_8 | 0.00 |
| 8 | inventory/wood_9 | 0.90 | 53 | inventory/sapphire_9 | 0.00 |
| 9 | inventory/stone_1 | 0.96 | 54 | inventory/ruby_1 | 0.01 |
| 10 | inventory/stone_2 | 0.96 | 55 | inventory/ruby_2 | 0.00 |
| 11 | inventory/stone_3 | 0.96 | 56 | inventory/ruby_3 | 0.00 |
| 12 | inventory/stone_4 | 0.96 | 57 | inventory/ruby_4 | 0.00 |
| 13 | inventory/stone_5 | 0.95 | 58 | inventory/ruby_5 | 0.00 |
| 14 | inventory/stone_6 | 0.95 | 59 | inventory/ruby_6 | 0.00 |
| 15 | inventory/stone_7 | 0.94 | 60 | inventory/ruby_7 | 0.00 |
| 16 | inventory/stone_8 | 0.93 | 61 | inventory/ruby_8 | 0.00 |
| 17 | inventory/stone_9 | 0.91 | 62 | inventory/ruby_9 | 0.00 |
| 18 | inventory/coal_1 | 0.57 | 63 | inventory/sapling_1 | 1.00 |
| 19 | inventory/coal_2 | 0.23 | 64 | inventory/sapling_2 | 1.00 |
| 20 | inventory/coal_3 | 0.07 | 65 | inventory/sapling_3 | 1.00 |
| 21 | inventory/coal_4 | 0.01 | 66 | inventory/sapling_4 | 1.00 |
| 22 | inventory/coal_5 | 0.00 | 67 | inventory/sapling_5 | 1.00 |
| 23 | inventory/coal_6 | 0.00 | 68 | inventory/sapling_6 | 1.00 |
| 24 | inventory/coal_7 | 0.00 | 69 | inventory/sapling_7 | 0.99 |
| 25 | inventory/coal_8 | 0.00 | 70 | inventory/sapling_8 | 0.99 |
| 26 | inventory/coal_9 | 0.00 | 71 | inventory/sapling_9 | 0.98 |
| 27 | inventory/iron_1 | 0.01 | 72 | inventory/torches_1 | 0.21 |
| 28 | inventory/iron_2 | 0.02 | 73 | inventory/torches_2 | 0.27 |
| 29 | inventory/iron_3 | 0.00 | 74 | inventory/torches_3 | 0.40 |
| 30 | inventory/iron_4 | 0.00 | 75 | inventory/torches_4 | 0.41 |
| 31 | inventory/iron_5 | 0.00 | 76 | inventory/torches_5 | 0.05 |
| 32 | inventory/iron_6 | 0.00 | 77 | inventory/torches_6 | 0.03 |
| 33 | inventory/iron_7 | 0.00 | 78 | inventory/torches_7 | 0.03 |
| 34 | inventory/iron_8 | 0.00 | 79 | inventory/torches_8 | 0.00 |
| 35 | inventory/iron_9 | 0.00 | 80 | inventory/torches_9 | 0.00 |
| 36 | inventory/diamond_1 | 0.01 | 81 | inventory/arrows_1 | 0.01 |
| 37 | inventory/diamond_2 | 0.00 | 82 | inventory/arrows_2 | 0.81 |
| 38 | inventory/diamond_3 | 0.00 | 83 | inventory/arrows_3 | 0.01 |
| 39 | inventory/diamond_4 | 0.00 | 84 | inventory/arrows_4 | 0.69 |
| 40 | inventory/diamond_5 | 0.00 | 85 | inventory/arrows_5 | 0.00 |
| 41 | inventory/diamond_6 | 0.00 | 86 | inventory/arrows_6 | 0.14 |
| 42 | inventory/diamond_7 | 0.00 | 87 | inventory/arrows_7 | 0.00 |
| 43 | inventory/diamond_8 | 0.00 | 88 | inventory/arrows_8 | 0.04 |
| 44 | inventory/diamond_9 | 0.00 | 89 | inventory/arrows_9 | 0.00 |

Table 3. Craftax goal listing. To give an indication of goal difficulty, we show the success rate of Dual LEO (PQN) after 1 billion timesteps, averaged over 5 seeds. Part 1.

| ID | Name | Success Rate | ID | Name | Success Rate |
|---|---|---|---|---|---|
| 100 | inventory/wood_60-64 | 0.00 | 150 | inventory/iron_40-44 | 0.00 |
| 101 | inventory/wood_65-69 | 0.00 | 151 | inventory/iron_45-49 | 0.00 |
| 102 | inventory/wood_70-74 | 0.00 | 152 | inventory/iron_50-54 | 0.00 |
| 103 | inventory/wood_75-79 | 0.00 | 153 | inventory/iron_55-59 | 0.00 |
| 104 | inventory/wood_80-84 | 0.00 | 154 | inventory/iron_60-64 | 0.00 |
| 105 | inventory/wood_85-89 | 0.00 | 155 | inventory/iron_65-69 | 0.00 |
| 106 | inventory/wood_90-94 | 0.00 | 156 | inventory/iron_70-74 | 0.00 |
| 107 | inventory/wood_95-99 | 0.00 | 157 | inventory/iron_75-79 | 0.00 |
| 108 | inventory/stone_10-14 | 0.87 | 158 | inventory/iron_80-84 | 0.00 |
| 109 | inventory/stone_15-19 | 0.24 | 159 | inventory/iron_85-89 | 0.00 |
| 110 | inventory/stone_20-24 | 0.00 | 160 | inventory/iron_90-94 | 0.00 |
| 111 | inventory/stone_25-29 | 0.00 | 161 | inventory/iron_95-99 | 0.00 |
| 112 | inventory/stone_30-34 | 0.00 | 162 | inventory/torches_10-14 | 0.00 |
| 113 | inventory/stone_35-39 | 0.00 | 163 | inventory/torches_15-19 | 0.00 |
| 114 | inventory/stone_40-44 | 0.00 | 164 | inventory/torches_20-24 | 0.00 |
| 115 | inventory/stone_45-49 | 0.00 | 165 | inventory/torches_25-29 | 0.00 |
| 116 | inventory/stone_50-54 | 0.00 | 166 | inventory/torches_30-34 | 0.00 |
| 117 | inventory/stone_55-59 | 0.00 | 167 | inventory/torches_35-39 | 0.00 |
| 118 | inventory/stone_60-64 | 0.00 | 168 | inventory/torches_40-44 | 0.00 |
| 119 | inventory/stone_65-69 | 0.00 | 169 | inventory/torches_45-49 | 0.00 |
| 120 | inventory/stone_70-74 | 0.00 | 170 | inventory/torches_50-54 | 0.00 |
| 121 | inventory/stone_75-79 | 0.00 | 171 | inventory/torches_55-59 | 0.00 |
| 122 | inventory/stone_80-84 | 0.00 | 172 | inventory/torches_60-64 | 0.00 |
| 123 | inventory/stone_85-89 | 0.00 | 173 | inventory/torches_65-69 | 0.00 |
| 124 | inventory/stone_90-94 | 0.00 | 174 | inventory/torches_70-74 | 0.00 |
| 125 | inventory/stone_95-99 | 0.00 | 175 | inventory/torches_75-79 | 0.00 |
| 126 | inventory/coal_10-14 | 0.00 | 176 | inventory/torches_80-84 | 0.00 |
| 127 | inventory/coal_15-19 | 0.00 | 177 | inventory/torches_85-89 | 0.00 |
| 128 | inventory/coal_20-24 | 0.00 | 178 | inventory/torches_90-94 | 0.00 |
| 129 | inventory/coal_25-29 | 0.00 | 179 | inventory/torches_95-99 | 0.00 |
| 130 | inventory/coal_30-34 | 0.00 | 180 | inventory/arrows_10-14 | 0.01 |
| 131 | inventory/coal_35-39 | 0.00 | 181 | inventory/arrows_15-19 | 0.00 |
| 132 | inventory/coal_40-44 | 0.00 | 182 | inventory/arrows_20-24 | 0.00 |
| 133 | inventory/coal_45-49 | 0.00 | 183 | inventory/arrows_25-29 | 0.00 |
| 134 | inventory/coal_50-54 | 0.00 | 184 | inventory/arrows_30-34 | 0.00 |
| 135 | inventory/coal_55-59 | 0.00 | 185 | inventory/arrows_35-39 | 0.00 |
| 136 | inventory/coal_60-64 | 0.00 | 186 | inventory/arrows_40-44 | 0.00 |
| 137 | inventory/coal_65-69 | 0.00 | 187 | inventory/arrows_45-49 | 0.00 |
| 138 | inventory/coal_70-74 | 0.00 | 188 | inventory/arrows_50-54 | 0.00 |
| 139 | inventory/coal_75-79 | 0.00 | 189 | inventory/arrows_55-59 | 0.00 |
| 140 | inventory/coal_80-84 | 0.00 | 190 | inventory/arrows_60-64 | 0.00 |
| 141 | inventory/coal_85-89 | 0.00 | 191 | inventory/arrows_65-69 | 0.00 |
| 142 | inventory/coal_90-94 | 0.00 | 192 | inventory/arrows_70-74 | 0.00 |
| 143 | inventory/coal_95-99 | 0.00 | 193 | inventory/arrows_75-79 | 0.00 |
| 144 | inventory/iron_10-14 | 0.00 | 194 | inventory/arrows_80-84 | 0.00 |
| 145 | inventory/iron_15-19 | 0.00 | 195 | inventory/arrows_85-89 | 0.00 |
| 146 | inventory/iron_20-24 | 0.00 | 196 | inventory/arrows_90-94 | 0.00 |
| 147 | inventory/iron_25-29 | 0.00 | 197 | inventory/arrows_95-99 | 0.00 |
| 148 | inventory/iron_30-34 | 0.00 | 198 | block_map/out_of_bounds_left | 0.00 |
| 149 | inventory/iron_35-39 | 0.00 | 199 | block_map/out_of_bounds_right | 0.00 |

*Table 4.* Craftax goal listing. To give an indication of goal difficulty, we show the success rate of Dual LEO (PQN) after 1 billion timesteps, averaged over 5 seeds. Part 2.

| ID | Name | Success Rate | ID | Name | Success Rate |
|---|---|---|---|---|---|
| 200 | block_map/out_of_bounds_up | 0.00 | 250 | block_map/plant_left | 1.00 |
| 201 | block_map/out_of_bounds_down | 0.00 | 251 | block_map/plant_right | 1.00 |
| 202 | block_map/grass_left | 1.00 | 252 | block_map/plant_up | 1.00 |
| 203 | block_map/grass_right | 1.00 | 253 | block_map/plant_down | 1.00 |
| 204 | block_map/grass_up | 1.00 | 254 | block_map/ripe_plant_left | 0.00 |
| 205 | block_map/grass_down | 1.00 | 255 | block_map/ripe_plant_right | 0.00 |
| 206 | block_map/water_left | 0.78 | 256 | block_map/ripe_plant_up | 0.00 |
| 207 | block_map/water_right | 0.79 | 257 | block_map/ripe_plant_down | 0.00 |
| 208 | block_map/water_up | 0.79 | 258 | block_map/wall_left | 0.19 |
| 209 | block_map/water_down | 0.77 | 259 | block_map/wall_right | 0.19 |
| 210 | block_map/stone_left | 0.95 | 260 | block_map/wall_up | 0.19 |
| 211 | block_map/stone_right | 0.95 | 261 | block_map/wall_down | 0.19 |
| 212 | block_map/stone_up | 0.95 | 262 | block_map/wall_moss_left | 0.08 |
| 213 | block_map/stone_down | 0.94 | 263 | block_map/wall_moss_right | 0.07 |
| 214 | block_map/tree_left | 0.99 | 264 | block_map/wall_moss_up | 0.10 |
| 215 | block_map/tree_right | 1.00 | 265 | block_map/wall_moss_down | 0.08 |
| 216 | block_map/tree_up | 1.00 | 266 | block_map/stalagmite_left | 0.00 |
| 217 | block_map/tree_down | 1.00 | 267 | block_map/stalagmite_right | 0.00 |
| 218 | block_map/path_left | 0.99 | 268 | block_map/stalagmite_up | 0.00 |
| 219 | block_map/path_right | 0.99 | 269 | block_map/stalagmite_down | 0.00 |
| 220 | block_map/path_up | 0.99 | 270 | block_map/sapphire_left | 0.00 |
| 221 | block_map/path_down | 0.99 | 271 | block_map/sapphire_right | 0.00 |
| 222 | block_map/coal_left | 0.32 | 272 | block_map/sapphire_up | 0.00 |
| 223 | block_map/coal_right | 0.34 | 273 | block_map/sapphire_down | 0.00 |
| 224 | block_map/coal_up | 0.34 | 274 | block_map/ruby_left | 0.00 |
| 225 | block_map/coal_down | 0.37 | 275 | block_map/ruby_right | 0.00 |
| 226 | block_map/iron_left | 0.22 | 276 | block_map/ruby_up | 0.00 |
| 227 | block_map/iron_right | 0.23 | 277 | block_map/ruby_down | 0.00 |
| 228 | block_map/iron_up | 0.24 | 278 | block_map/chest_left | 0.15 |
| 229 | block_map/iron_down | 0.24 | 279 | block_map/chest_right | 0.15 |
| 230 | block_map/diamond_left | 0.01 | 280 | block_map/chest_up | 0.16 |
| 231 | block_map/diamond_right | 0.02 | 281 | block_map/chest_down | 0.15 |
| 232 | block_map/diamond_up | 0.02 | 282 | block_map/fountain_left | 0.03 |
| 233 | block_map/diamond_down | 0.01 | 283 | block_map/fountain_right | 0.03 |
| 234 | block_map/crafting_table_left | 0.99 | 284 | block_map/fountain_up | 0.04 |
| 235 | block_map/crafting_table_right | 0.99 | 285 | block_map/fountain_down | 0.03 |
| 236 | block_map/crafting_table_up | 0.99 | 286 | block_map/fire_grass_left | 0.00 |
| 237 | block_map/crafting_table_down | 0.99 | 287 | block_map/fire_grass_right | 0.00 |
| 238 | block_map/furnace_left | 0.95 | 288 | block_map/fire_grass_up | 0.00 |
| 239 | block_map/furnace_right | 0.95 | 289 | block_map/fire_grass_down | 0.00 |
| 240 | block_map/furnace_up | 0.95 | 290 | block_map/ice_grass_left | 0.00 |
| 241 | block_map/furnace_down | 0.95 | 291 | block_map/ice_grass_right | 0.00 |
| 242 | block_map/sand_left | 0.96 | 292 | block_map/ice_grass_up | 0.00 |
| 243 | block_map/sand_right | 0.97 | 293 | block_map/ice_grass_down | 0.00 |
| 244 | block_map/sand_up | 0.96 | 294 | block_map/fire_tree_left | 0.00 |
| 245 | block_map/sand_down | 0.97 | 295 | block_map/fire_tree_right | 0.00 |
| 246 | block_map/lava_left | 0.37 | 296 | block_map/fire_tree_up | 0.00 |
| 247 | block_map/lava_right | 0.38 | 297 | block_map/fire_tree_down | 0.00 |
| 248 | block_map/lava_up | 0.34 | 298 | block_map/ice_shrub_left | 0.00 |
| 249 | block_map/lava_down | 0.35 | 299 | block_map/ice_shrub_right | 0.00 |

*Table 5.* Craftax goal listing. To give an indication of goal difficulty, we show the success rate of Dual LEO (PQN) after 1 billion timesteps, averaged over 5 seeds. Part 3.

| ID | Name | Success Rate | ID | Name | Success Rate |
|---|---|---|---|---|---|
| 300 | block_map/ice_shrub_up | 0.00 | 350 | mob_map/troll_left | 0.00 |
| 301 | block_map/ice_shrub_down | 0.00 | 351 | mob_map/troll_right | 0.00 |
| 302 | block_map/enchanter_fire_left | 0.00 | 352 | mob_map/troll_up | 0.00 |
| 303 | block_map/enchanter_fire_right | 0.00 | 353 | mob_map/troll_down | 0.00 |
| 304 | block_map/enchanter_fire_up | 0.00 | 354 | mob_map/pigman_left | 0.00 |
| 305 | block_map/enchanter_fire_down | 0.00 | 355 | mob_map/pigman_right | 0.00 |
| 306 | block_map/enchanter_ice_left | 0.00 | 356 | mob_map/pigman_up | 0.00 |
| 307 | block_map/enchanter_ice_right | 0.00 | 357 | mob_map/pigman_down | 0.00 |
| 308 | block_map/enchanter_ice_up | 0.00 | 358 | mob_map/frost_troll_left | 0.00 |
| 309 | block_map/enchanter_ice_down | 0.00 | 359 | mob_map/frost_troll_right | 0.00 |
| 310 | block_map/necromancer_left | 0.00 | 360 | mob_map/frost_troll_up | 0.00 |
| 311 | block_map/necromancer_right | 0.00 | 361 | mob_map/frost_troll_down | 0.00 |
| 312 | block_map/necromancer_up | 0.00 | 362 | mob_map/cow_left | 0.99 |
| 313 | block_map/necromancer_down | 0.00 | 363 | mob_map/cow_right | 0.99 |
| 314 | block_map/grave_left | 0.00 | 364 | mob_map/cow_up | 0.99 |
| 315 | block_map/grave_right | 0.00 | 365 | mob_map/cow_down | 0.99 |
| 316 | block_map/grave_up | 0.00 | 366 | mob_map/bat_left | 0.00 |
| 317 | block_map/grave_down | 0.00 | 367 | mob_map/bat_right | 0.00 |
| 318 | block_map/grave2_left | 0.00 | 368 | mob_map/bat_up | 0.00 |
| 319 | block_map/grave2_right | 0.00 | 369 | mob_map/bat_down | 0.00 |
| 320 | block_map/grave2_up | 0.00 | 370 | mob_map/snail_left | 0.12 |
| 321 | block_map/grave2_down | 0.00 | 371 | mob_map/snail_right | 0.10 |
| 322 | block_map/grave3_left | 0.00 | 372 | mob_map/snail_up | 0.10 |
| 323 | block_map/grave3_right | 0.00 | 373 | mob_map/snail_down | 0.11 |
| 324 | block_map/grave3_up | 0.00 | 374 | mob_map/skeleton_left | 0.69 |
| 325 | block_map/grave3_down | 0.00 | 375 | mob_map/skeleton_right | 0.68 |
| 326 | block_map/necromancer_hurt_left | 0.00 | 376 | mob_map/skeleton_up | 0.69 |
| 327 | block_map/necromancer_hurt_right | 0.00 | 377 | mob_map/skeleton_down | 0.68 |
| 328 | block_map/necromancer_hurt_up | 0.00 | 378 | mob_map/gnome_archer_left | 0.00 |
| 329 | block_map/necromancer_hurt_down | 0.00 | 379 | mob_map/gnome_archer_right | 0.00 |
| 330 | mob_map/zombie_left | 0.67 | 380 | mob_map/gnome_archer_up | 0.00 |
| 331 | mob_map/zombie_right | 0.69 | 381 | mob_map/gnome_archer_down | 0.00 |
| 332 | mob_map/zombie_up | 0.67 | 382 | mob_map/orc_mage_left | 0.03 |
| 333 | mob_map/zombie_down | 0.66 | 383 | mob_map/orc_mage_right | 0.02 |
| 334 | mob_map/gnome_warrior_left | 0.00 | 384 | mob_map/orc_mage_up | 0.02 |
| 335 | mob_map/gnome_warrior_right | 0.00 | 385 | mob_map/orc_mage_down | 0.02 |
| 336 | mob_map/gnome_warrior_up | 0.00 | 386 | mob_map/kobold_left | 0.00 |
| 337 | mob_map/gnome_warrior_down | 0.00 | 387 | mob_map/kobold_right | 0.00 |
| 338 | mob_map/orc_soldier_left | 0.11 | 388 | mob_map/kobold_up | 0.00 |
| 339 | mob_map/orc_soldier_right | 0.10 | 389 | mob_map/kobold_down | 0.00 |
| 340 | mob_map/orc_soldier_up | 0.11 | 390 | mob_map/knight_archer_left | 0.00 |
| 341 | mob_map/orc_soldier_down | 0.10 | 391 | mob_map/knight_archer_right | 0.00 |
| 342 | mob_map/lizard_left | 0.00 | 392 | mob_map/knight_archer_up | 0.00 |
| 343 | mob_map/lizard_right | 0.00 | 393 | mob_map/knight_archer_down | 0.00 |
| 344 | mob_map/lizard_up | 0.00 | 394 | mob_map/deep_thing_left | 0.00 |
| 345 | mob_map/lizard_down | 0.00 | 395 | mob_map/deep_thing_right | 0.00 |
| 346 | mob_map/knight_left | 0.00 | 396 | mob_map/deep_thing_up | 0.00 |
| 347 | mob_map/knight_right | 0.00 | 397 | mob_map/deep_thing_down | 0.00 |
| 348 | mob_map/knight_up | 0.00 | 398 | mob_map/fire_elemental_left | 0.00 |
| 349 | mob_map/knight_down | 0.00 | 399 | mob_map/fire_elemental_right | 0.00 |

*Table 6.* Craftax goal listing. To give an indication of goal difficulty, we show the success rate of Dual LEO (PQN) after 1 billion timesteps, averaged over 5 seeds. Part 4.

| ID | Name | Success Rate | ID | Name | Success Rate |
|----|------|--------------|----|------|--------------|
| 400 | mob_map/fire_elemental_up | 0.00 | 450 | item_map/ladder_down_left | 0.16 |
| 401 | mob_map/fire_elemental_down | 0.00 | 451 | item_map/ladder_down_right | 0.16 |
| 402 | mob_map/ice_elemental_left | 0.00 | 452 | item_map/ladder_down_up | 0.16 |
| 403 | mob_map/ice_elemental_right | 0.00 | 453 | item_map/ladder_down_down | 0.16 |
| 404 | mob_map/ice_elemental_up | 0.00 | 454 | item_map/ladder_up_left | 0.16 |
| 405 | mob_map/ice_elemental_down | 0.00 | 455 | item_map/ladder_up_right | 0.14 |
| 406 | mob_map/arrow_left | 0.68 | 456 | item_map/ladder_up_up | 0.14 |
| 407 | mob_map/arrow_right | 0.68 | 457 | item_map/ladder_up_down | 0.15 |
| 408 | mob_map/arrow_up | 0.70 | 458 | item_map/ladder_down_blocked_left | 0.00 |
| 409 | mob_map/arrow_down | 0.68 | 459 | item_map/ladder_down_blocked_right | 0.00 |
| 410 | mob_map/dagger_left | 0.00 | 460 | item_map/ladder_down_blocked_up | 0.00 |
| 411 | mob_map/dagger_right | 0.00 | 461 | item_map/ladder_down_blocked_down | 0.00 |
| 412 | mob_map/dagger_up | 0.00 | 462 | tools/wood_pickaxe | 0.98 |
| 413 | mob_map/dagger_down | 0.00 | 463 | tools/wood_sword | 0.98 |
| 414 | mob_map/fireball_left | 0.05 | 464 | tools/stone_pickaxe | 0.87 |
| 415 | mob_map/fireball_right | 0.05 | 465 | tools/stone_sword | 0.88 |
| 416 | mob_map/fireball_up | 0.07 | 466 | tools/iron_pickaxe | 0.00 |
| 417 | mob_map/fireball_down | 0.05 | 467 | tools/iron_sword | 0.00 |
| 418 | mob_map/arrow2_left | 0.00 | 468 | tools/diamond_pickaxe | 0.00 |
| 419 | mob_map/arrow2_right | 0.00 | 469 | tools/diamond_sword | 0.00 |
| 420 | mob_map/arrow2_up | 0.00 | 470 | tools/bow | 0.15 |
| 421 | mob_map/arrow2_down | 0.00 | 471 | tools/iron_helmet | 0.00 |
| 422 | mob_map/slimeball_left | 0.00 | 472 | tools/diamond_helmet | 0.00 |
| 423 | mob_map/slimeball_right | 0.00 | 473 | tools/iron_chestplate | 0.00 |
| 424 | mob_map/slimeball_up | 0.00 | 474 | tools/diamond_chestplate | 0.00 |
| 425 | mob_map/slimeball_down | 0.00 | 475 | tools/iron_pants | 0.00 |
| 426 | mob_map/fireball2_left | 0.00 | 476 | tools/diamond_pants | 0.00 |
| 427 | mob_map/fireball2_right | 0.00 | 477 | tools/iron_boots | 0.00 |
| 428 | mob_map/fireball2_up | 0.00 | 478 | tools/diamond_boots | 0.00 |
| 429 | mob_map/fireball2_down | 0.00 | 479 | enchant/helmet_fire | 0.00 |
| 430 | mob_map/iceball2_left | 0.00 | 480 | enchant/helmet_ice | 0.00 |
| 431 | mob_map/iceball2_right | 0.00 | 481 | enchant/chestplate_fire | 0.00 |
| 432 | mob_map/iceball2_up | 0.00 | 482 | enchant/chestplate_ice | 0.00 |
| 433 | mob_map/iceball2_down | 0.00 | 483 | enchant/pants_fire | 0.00 |
| 434 | mob_map/player_fireball_left | 0.00 | 484 | enchant/pants_ice | 0.00 |
| 435 | mob_map/player_fireball_right | 0.00 | 485 | enchant/boots_fire | 0.00 |
| 436 | mob_map/player_fireball_up | 0.00 | 486 | enchant/boots_ice | 0.00 |
| 437 | mob_map/player_fireball_down | 0.00 | 487 | enchant/sword_fire | 0.00 |
| 438 | mob_map/player_iceball_left | 0.00 | 488 | enchant/bow_fire | 0.00 |
| 439 | mob_map/player_iceball_right | 0.00 | 489 | enchant/sword_ice | 0.00 |
| 440 | mob_map/player_iceball_up | 0.00 | 490 | enchant/bow_ice | 0.00 |
| 441 | mob_map/player_iceball_down | 0.00 | 491 | dungeon_level/dlvl_0 | 1.00 |
| 442 | mob_map/player_arrow_left | 0.00 | 492 | dungeon_level/dlvl_1 | 0.20 |
| 443 | mob_map/player_arrow_right | 0.00 | 493 | dungeon_level/dlvl_2 | 0.00 |
| 444 | mob_map/player_arrow_up | 0.00 | 494 | dungeon_level/dlvl_3 | 0.00 |
| 445 | mob_map/player_arrow_down | 0.00 | 495 | dungeon_level/dlvl_4 | 0.00 |
| 446 | item_map/torch_left | 0.30 | 496 | dungeon_level/dlvl_5 | 0.00 |
| 447 | item_map/torch_right | 0.32 | 497 | dungeon_level/dlvl_6 | 0.00 |
| 448 | item_map/torch_up | 0.31 | 498 | dungeon_level/dlvl_7 | 0.00 |
| 449 | item_map/torch_down | 0.32 | 499 | dungeon_level/dlvl_8 | 0.00 |

*Table 7.* Craftax goal listing. To give an indication of goal difficulty, we show the success rate of Dual LEO (PQN) after 1 billion timesteps, averaged over 5 seeds. Part 5.

| ID | Name | Success Rate |
|---|---|---|
| 500 | intrinsics/intelligence_2 | 0.21 |
| 501 | intrinsics/intelligence_3 | 0.00 |
| 502 | intrinsics/intelligence_4 | 0.00 |
| 503 | intrinsics/intelligence_5 | 0.00 |
| 504 | intrinsics/strength_2 | 0.19 |
| 505 | intrinsics/strength_3 | 0.00 |
| 506 | intrinsics/strength_4 | 0.00 |
| 507 | intrinsics/strength_5 | 0.00 |
| 508 | intrinsics/dexterity_2 | 0.19 |
| 509 | intrinsics/dexterity_3 | 0.00 |
| 510 | intrinsics/dexterity_4 | 0.00 |
| 511 | intrinsics/dexterity_5 | 0.00 |
| 90 | inventory/wood_10-14 | 0.90 |
| 91 | inventory/wood_15-19 | 0.07 |
| 92 | inventory/wood_20-24 | 0.00 |
| 93 | inventory/wood_25-29 | 0.00 |
| 94 | inventory/wood_30-34 | 0.00 |
| 95 | inventory/wood_35-39 | 0.00 |
| 96 | inventory/wood_40-44 | 0.00 |
| 97 | inventory/wood_45-49 | 0.00 |
| 98 | inventory/wood_50-54 | 0.00 |
| 99 | inventory/wood_55-59 | 0.00 |

*Table 8.* Craftax goal listing. To give an indication of goal difficulty, we show the success rate of Dual LEO (PQN) after 1 billion timesteps, averaged over 5 seeds. Part 6.

## C. CraftaxGC per-goal results

Results for all goals are shown in Figures 8, 9, 10, 11, 12, 13, 14 for Craftax and Figures 15 and 16 for Craftax-Classic. Note that, especially for Craftax, many goals are never achieved.

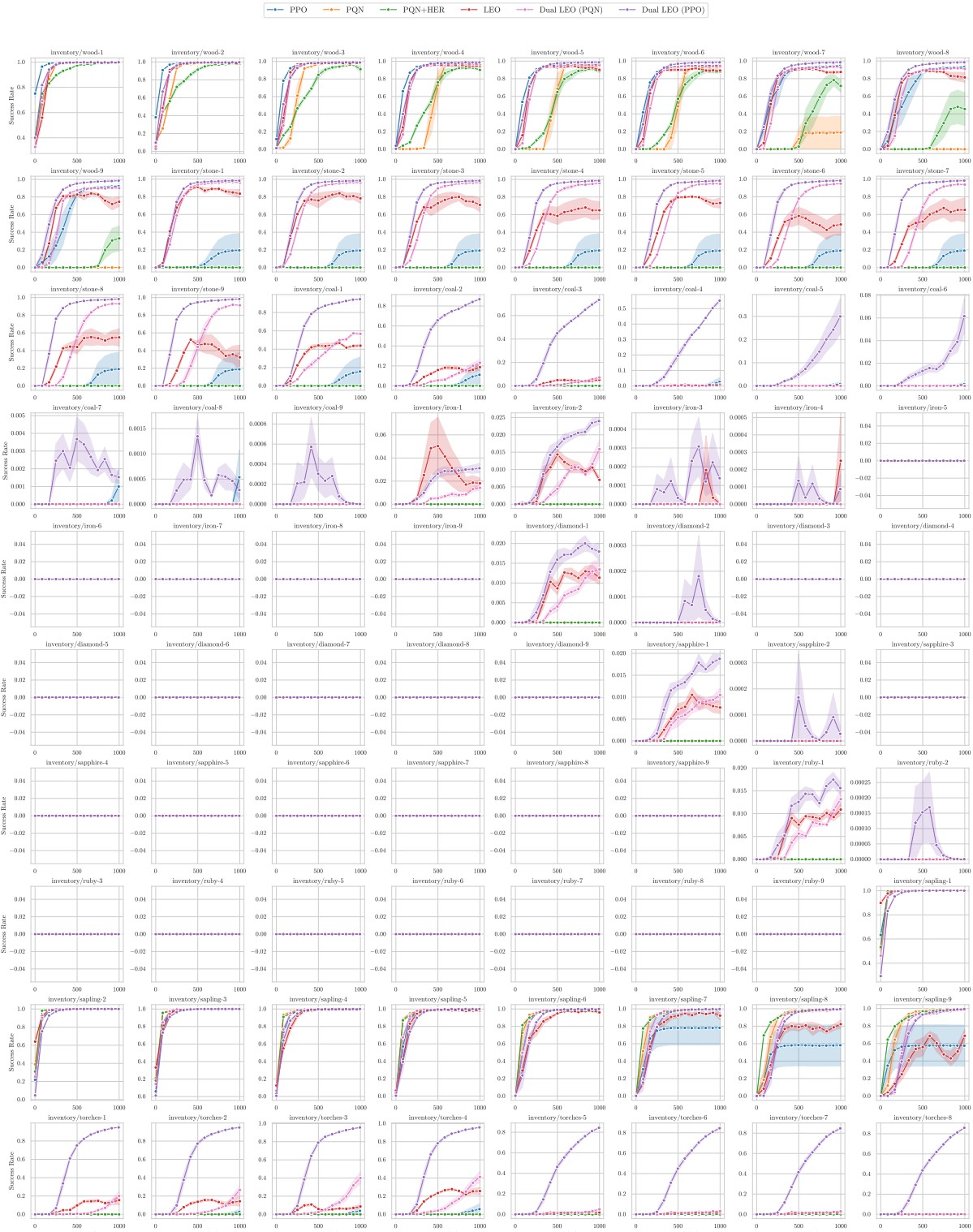

*Figure 8.* Mean success rates for all algorithms on CraftaxGC. Shaded area denotes 1 standard error over 5 seeds. Part 1.

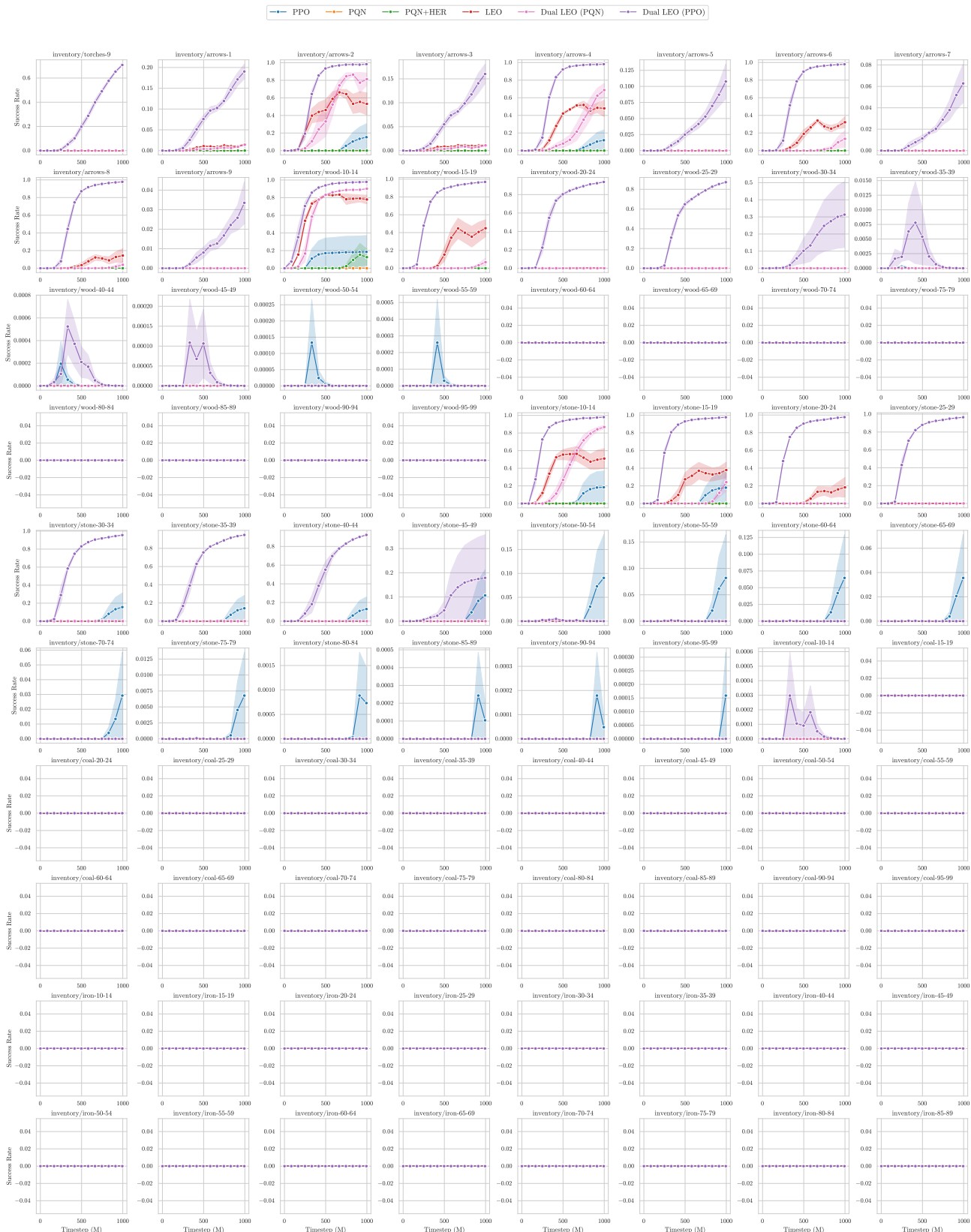

*Figure 9.* Mean success rates for all algorithms on CraftaxGC. Shaded area denotes 1 standard error over 5 seeds. Part 2.

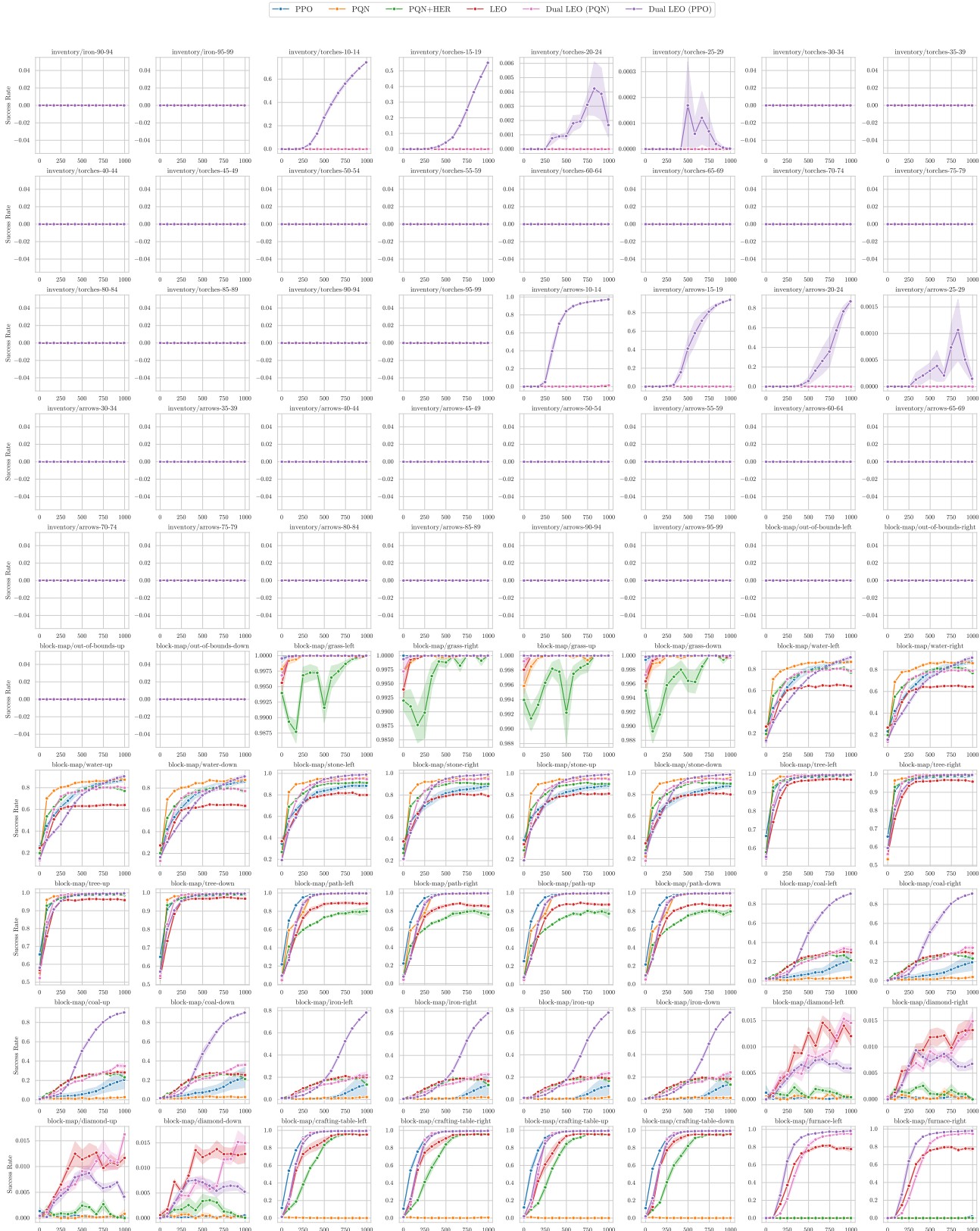

*Figure 10.* Mean success rates for all algorithms on CraftaxGC. Shaded area denotes 1 standard error over 5 seeds. Part 3.

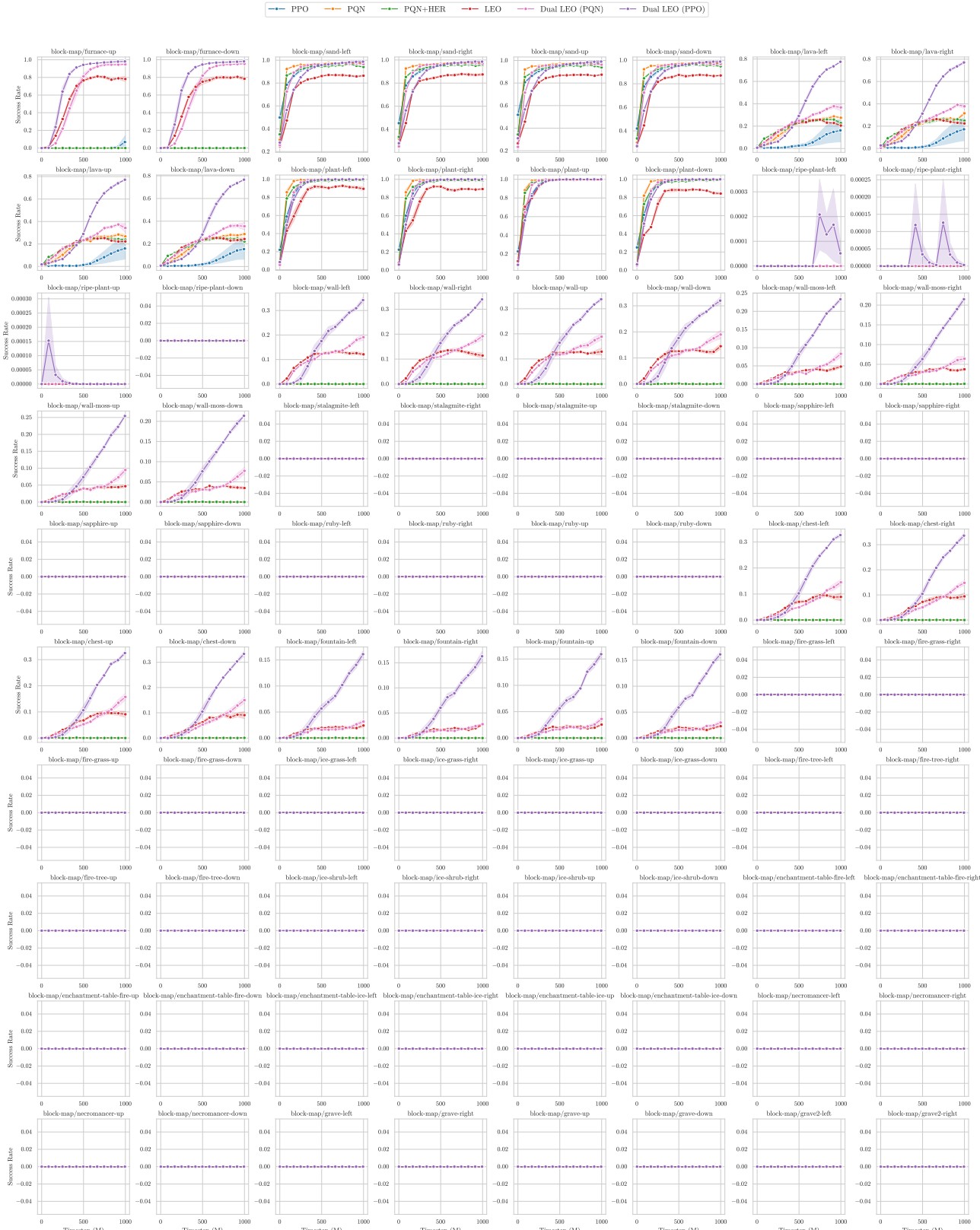

*Figure 11.* Mean success rates for all algorithms on CraftaxGC. Shaded area denotes 1 standard error over 5 seeds. Part 4.

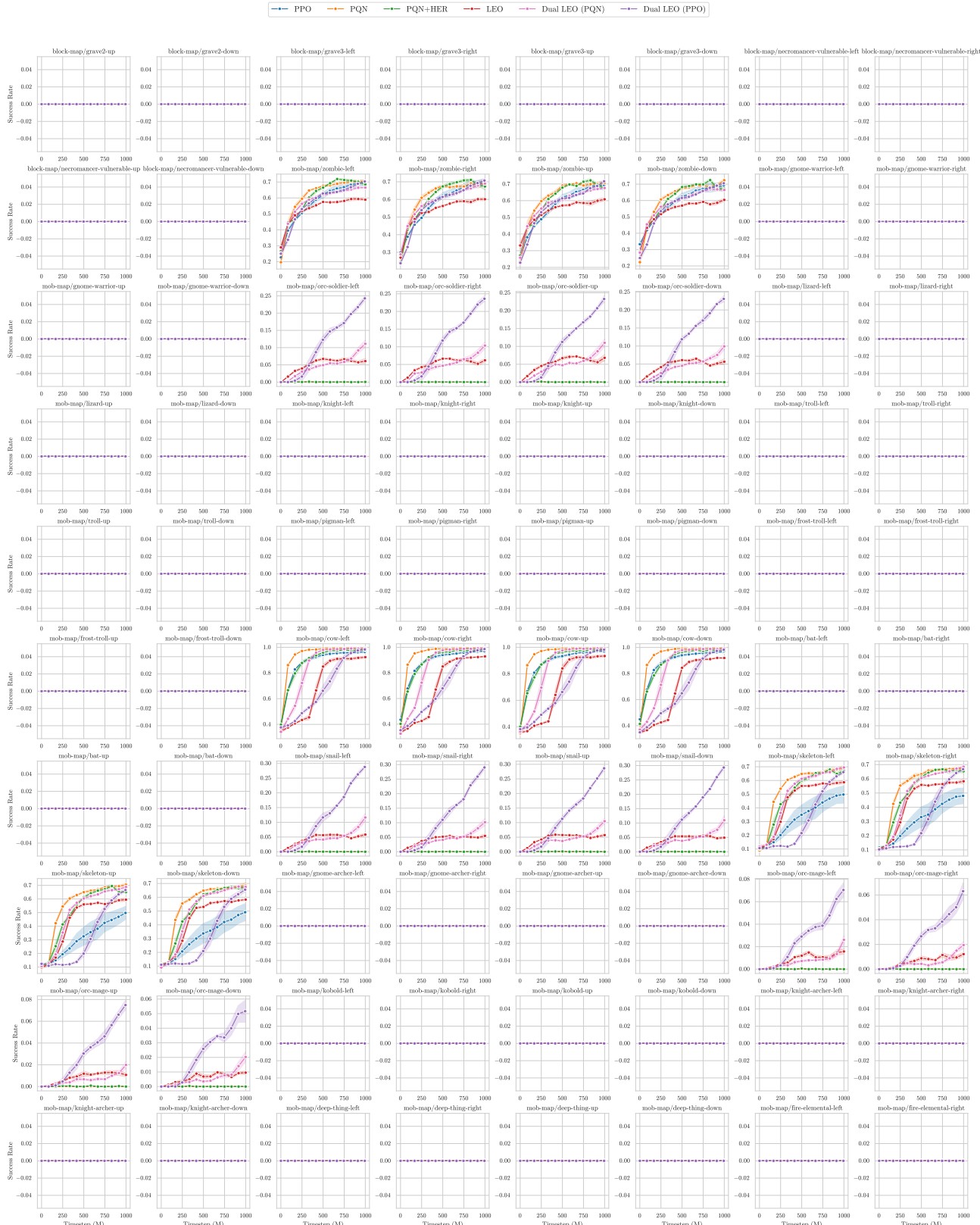

*Figure 12.* Mean success rates for all algorithms on CraftaxGC. Shaded area denotes 1 standard error over 5 seeds. Part 5.

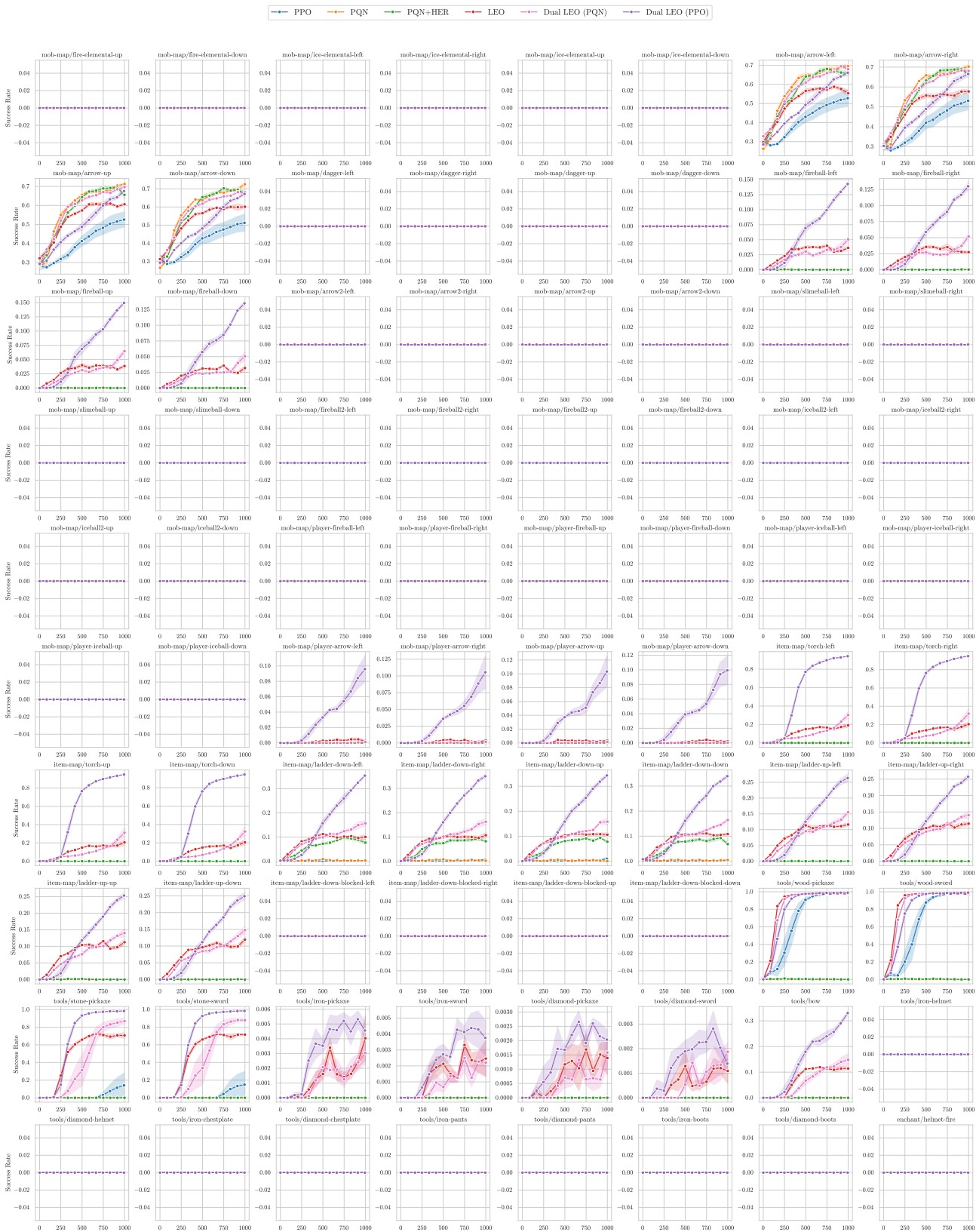

*Figure 13.* Mean success rates for all algorithms on CraftaxGC. Shaded area denotes 1 standard error over 5 seeds. Part 6.

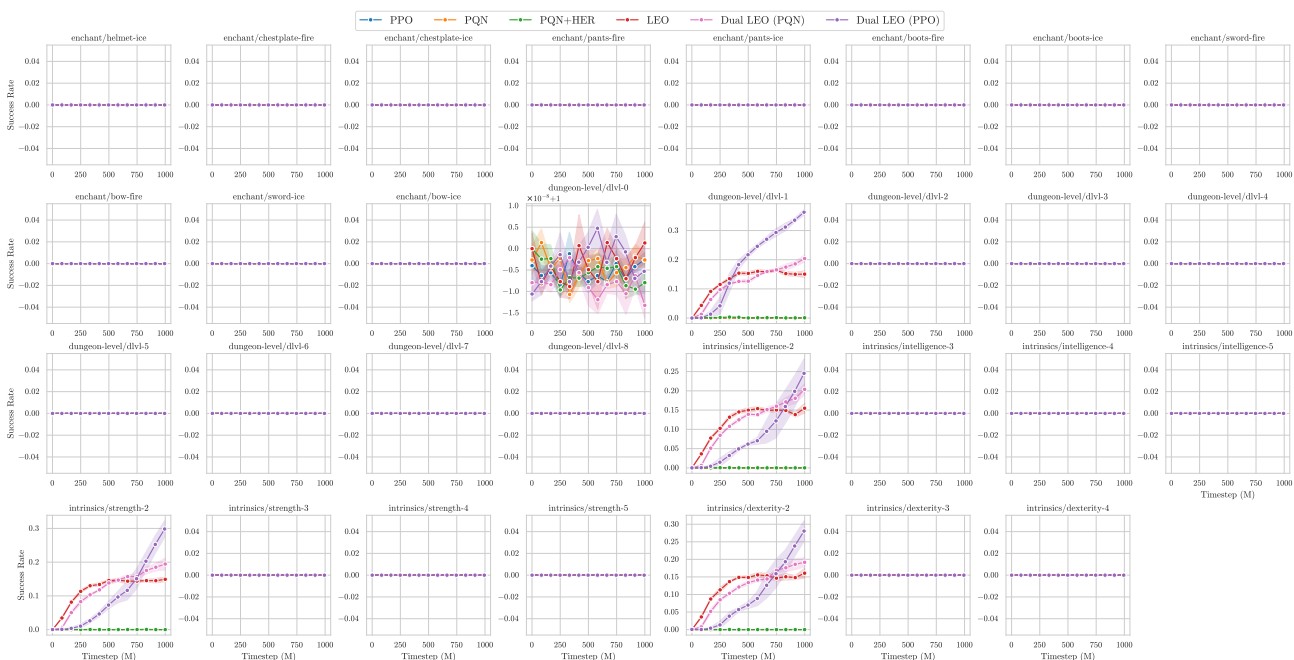

*Figure 14.* Mean success rates for all algorithms on CraftaxGC. Shaded area denotes 1 standard error over 5 seeds. Part 7.

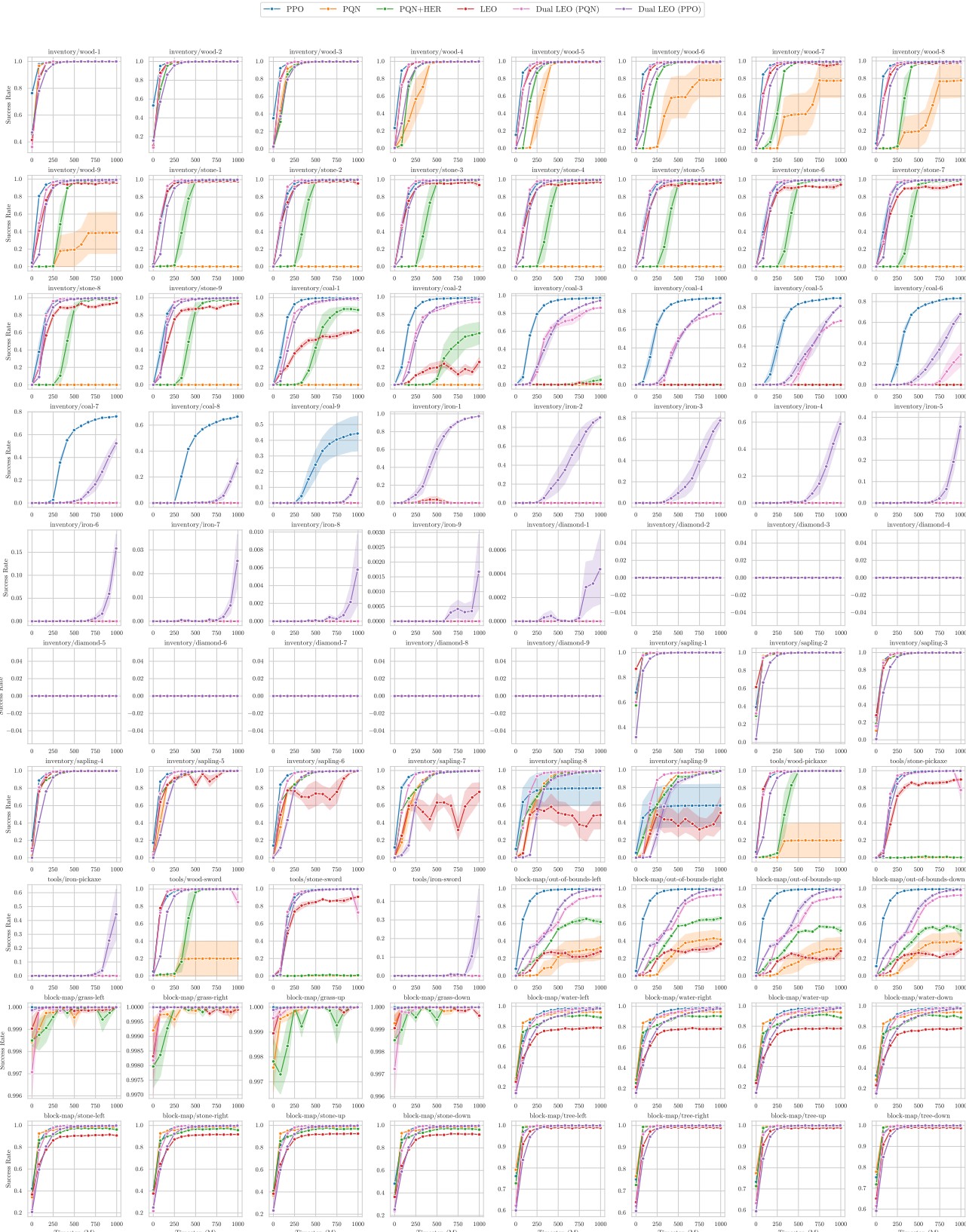

*Figure 15.* Mean success rates for all algorithms on CraftaxGC for Craftax-Classic. Shaded area denotes 1 standard error over 5 seeds. Part 1.

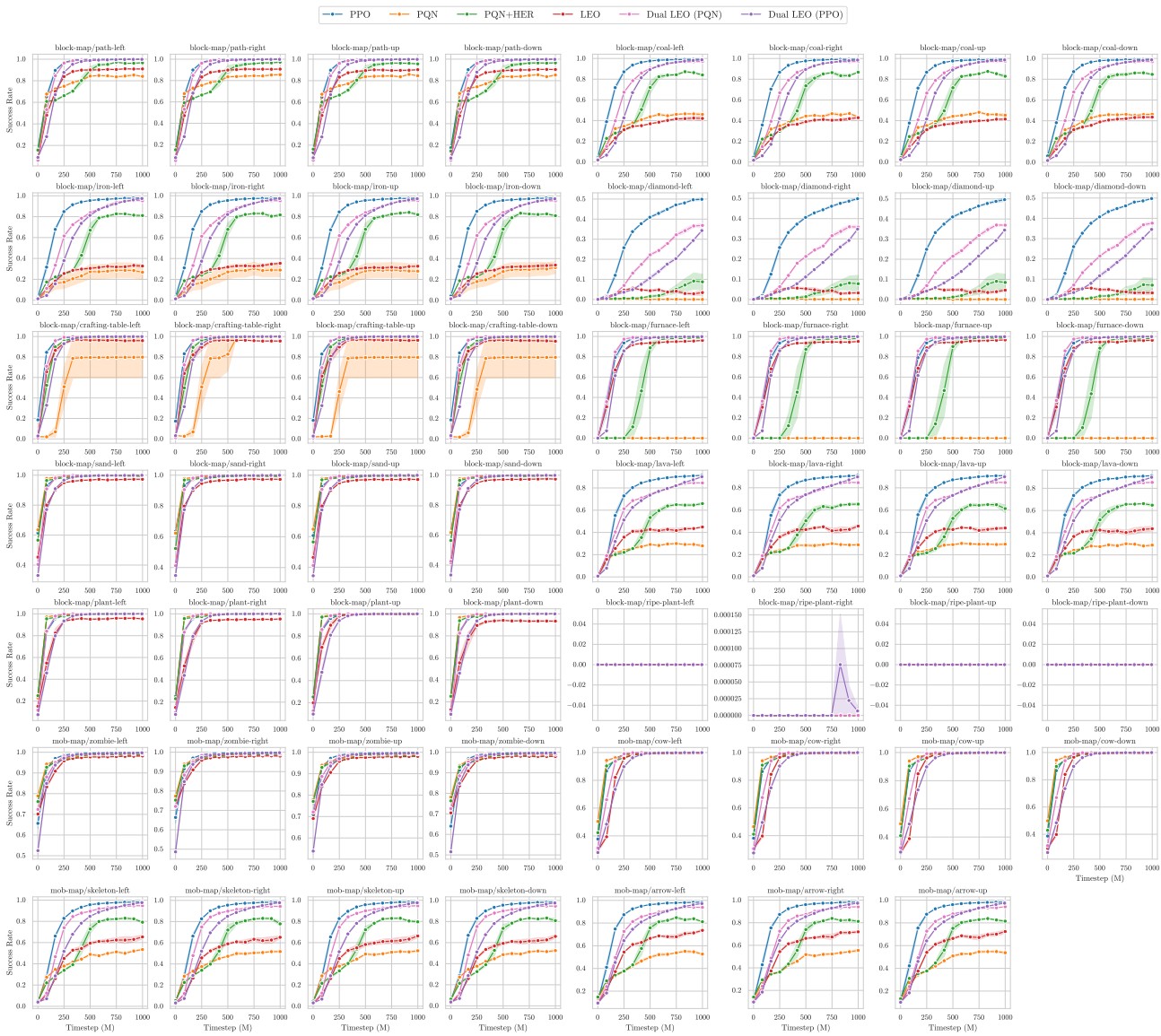

*Figure 16.* Mean success rates for all algorithms on CraftaxGC for Craftax-Classic. Shaded area denotes 1 standard error over 5 seeds. Part 2.

## D. CraftaxGC Hyperparameters

For PPO, PQN and LEO, we ran a sweep of 100 runs of 200 million timesteps, each with a uniformly sampled set of hyperparameters, with the results shown in Tables 9, 10 and 11. For Dual LEO (PQN) we took the best PQN hyperparameters and then additionally swept over the new hyperparameters, as shown in Table 12. For Dual LEO (PPO) we took the best PPO hyperparameters and then additionally swept over the new hyperparameters in Table 13.

| Hyperparameter | Values Considered | Best Value |
|---|---|---|
| Network Dense Hidden Size | $\{256, 512, 1024\}$ | 1024 |
| Network Embedding Layers | $\{1, 2, 4\}$ | 1 |
| Network Critic Layers | $\{1, 2, 4\}$ | 4 |
| Network Actor Layers | $\{1, 2, 4\}$ | 2 |
| Network Convolutional Features | $\{4, 8, 16, 32\}$ | 32 |
| # Steps | $\{1, 2, 4, 8, 16, 32, 64\}$ | 64 |
| # Epochs | $\{1, 2, 4\}$ | 1 |
| Learning Rate | $\{0.0002, 0.0003\}$ | 0.0002 |
| Learning Rate Decay | $\{$true, false$\}$ | true |
| $\gamma$ | $\{0.99, 0.995, 0.999\}$ | 0.995 |
| # Minibatches | $\{8, 16, 32\}$ | 32 |
| GAE $\lambda$ | $\{0.8, 0.9, 0.95\}$ | 0.95 |
| Entropy Coefficient | $\{0.01, 0.005, 0.002\}$ | 0.005 |
| Value Function Coefficient | $\{0.5\}$ | 0.5 |

*Table 9.* CraftaxGC PPO hyperparameter sweep. A random sweep of 100 runs for 200 million timesteps each was used to select the hyperparameters.

| Hyperparameter | Values Considered | Best Value |
|---|---|---|
| Network Dense Hidden Size | $\{256, 512, 1024\}$ | 1024 |
| Network Dense Layers | $\{1, 2, 4\}$ | 4 |
| Network Convolutional Features | $\{4, 8, 16, 32\}$ | 16 |
| # Steps | $\{1, 2, 4, 8, 16, 32, 64\}$ | 2 |
| Initial $\epsilon$ | $\{1.0, 0.5, 0.2, 0.1\}$ | 0.2 |
| Final $\epsilon$ | $\{0.005, 0.01, 0.02\}$ | 0.01 |
| $\epsilon$ Decay | $\{0.1, 0.2, 0.5\}$ | 0.5 |
| # Epochs | $\{1, 2, 4\}$ | 1 |
| Learning Rate | $\{0.0002, 0.0003\}$ | 0.0002 |
| Learning Rate Decay | $\{$true, false$\}$ | true |
| $\gamma$ | $\{0.99, 0.995, 0.999\}$ | 0.995 |
| Minibatch Size | $\{16, 32, 64, 128, 256, 512\}$ | 256 |

*Table 10.* CraftaxGC PQN hyperparameter sweep. A random sweep of 100 runs for 200 million timesteps each was used to select the hyperparameters.

| Hyperparameter | Values Considered | Best Value |
|---|---|---|
| Network Dense Hidden Size | $\{256, 512, 1024\}$ | 1024 |
| Network Dense Layers | $\{1, 2, 4, 6\}$ | 4 |
| Network Convolutional Features | $\{4, 8, 16, 32\}$ | 32 |
| # Steps | $\{1, 2, 4, 8, 16, 32, 64\}$ | 32 |
| Initial $\epsilon$ | $\{1.0, 0.5, 0.2, 0.1\}$ | 0.2 |
| Final $\epsilon$ | $\{0.005, 0.01, 0.02\}$ | 0.01 |
| $\epsilon$ Decay | $\{0.1, 0.2, 0.5\}$ | 0.2 |
| # Epochs | $\{1, 2, 4\}$ | 2 |
| Learning Rate | $\{0.0002, 0.0003\}$ | 0.0002 |
| Learning Rate Decay | $\{\text{true}, \text{false}\}$ | true |
| $\gamma$ | $\{0.99, 0.995, 0.999\}$ | 0.99 |
| Minibatch Size | $\{16, 32, 64, 128, 256, 512\}$ | 512 |

*Table 11.* CraftaxGC LEO hyperparameter sweep. A random sweep of 100 runs for 200 million timesteps each was used to select the hyperparameters.

| Hyperparameter | Values Considered | Best Value |
|---|---|---|
| Acting Method | $\{\text{max}, \text{min}, \text{linear combination}\}$ | linear combination |
| Linear Combination $\alpha$ | $\{0.1, 0.2, 0.3, 0.4, 0.5, 0.6, 0.7, 0.8, 0.9\}$ | 0.3 |
| Anneal $\alpha$ | $\{\text{true}, \text{false}\}$ | false |

*Table 12.* CraftaxGC Dual LEO (PQN) hyperparameter sweep.

| Hyperparameter | Values Considered | Best Value |
|---|---|---|
| Policy Cloning Coefficient | $\{0.0, 0.003, 0.01, 0.03, 0.1, 0.3, 1.0, 3.0\}$ | 0.1 |
| Value Cloning Coefficient | $\{0.0, 0.003, 0.01, 0.03, 0.1, 0.3, 1.0, 3.0\}$ | 0.0 |
| Anneal | $\{\text{true}, \text{false}\}$ | true |

*Table 13.* CraftaxGC Dual LEO (PPO) hyperparameter sweep.

# E. Hindsight Experience Replay Hyperparameters

We compare against various hindsight relabelling strategies:

- **None** No hindsight relabelling is performed.

- **Random ($n$)** We synthesise $n$ extra transitio'ns, relabelled with random goals.

- **Positive ($m$)** We synthesise $m$ extra transitions, relabelled with random goals that are achieved at each timestep. Note that in CraftaxGC, many goals are achieved every timestep.

- **Mixed ($n + m$)** We synthesise $n$ extra random transitions and $m$ extra positive transitions.

For each strategy we consider applying it to each transition individually, as well as obtaining the goals from the final transition in the batch and relabelling the entire sub-trajectory the same (as is done typically in HER). We run each setting on Craftax-Classic for 500M timesteps. For Random and Positive we try $\{1, 2, 4\}$ relabels and for Mixed we try the cross product of $\{2, 4, 8\}$ relabels and $\{0.25, 0.5, 0.75\}$ proportion of the mixed goals being random. We find that relabelling with Mixed $(1 + 1)$ over a trajectory level has the best performance, and use this for our comparisons.

# F. Craftax Results with Uniform Goal Sampling

Figure 17 shows the results on CraftaxGC when sampling uniformly from the goal distribution, rather than only from previously observed goals. This makes little difference on Craftax-Classic, but makes all methods perform significantly worse on Craftax. LEO and Dual LEO still perform well, with the gap between them and the baselines being even bigger, showing that they are more resilient to the difficulty of the goal distribution. LEO can still learn a considerable amount from an episode where an impossible/unachievable goal is commanded.

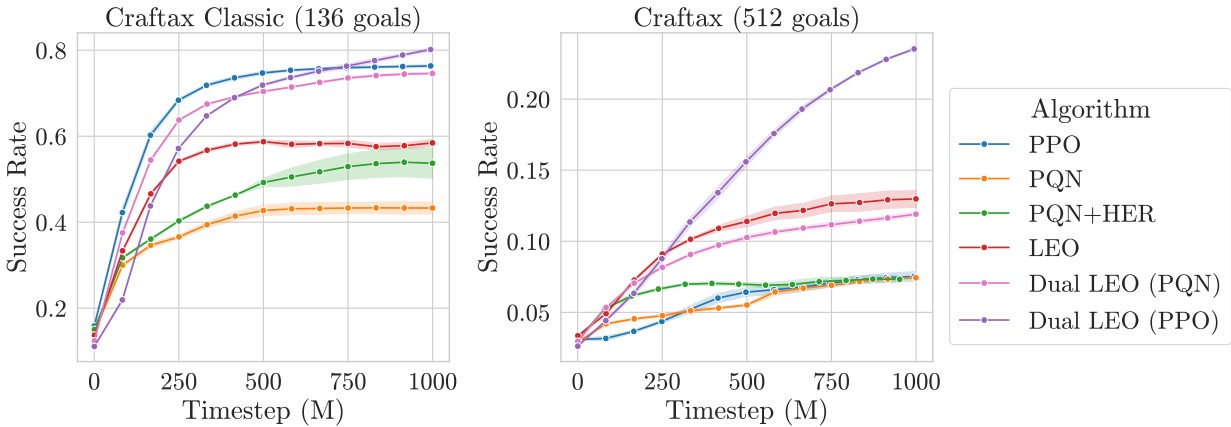

*Figure 17.* Results on CraftaxGC when sampling uniformly from the goal distribution, rather than only from seen goals. Shaded area denotes 1 standard error over 5 seeds.

# G. Subsampled Craftax Goal Set

We see from Figure 4 that PPO beats LEO for Craftax-Classic but loses on the full Craftax environment and goal set. To try and smoothly interpolate between these results, we run experiments on Craftax with randomly subsampled goal sets of varying sizes (Figure 18). We see that PPO monotonically trends downwards in performance as the size of the goal set increases. As each transition is only updated to with respect to a single goal, the *relative* learning capability of PPO with respect to the size of the goal set decreases. On the other hand, LEO stabilises for goal sets of size > 150 and performance does not decrease, as it can make use of the all-goals update.

While the downward trend holds for most goals, we do see some for which the opposite is true, such as the `tools/stone-pickaxe` goal (Figure 19). In our setup, both exploration and learning are intrinsically tied together. A smaller goal set is potentially easier to learn on with respect to a fixed dataset, but may lack 'stepping stone' goals that encourage exploration. For medium difficulty goals like `tools/stone-pickaxe`, it seems that the benefits of stepping stones outweigh the need to learn on a larger goal set. Note that in order to do this single-goal analysis we edited every subsampled goal set to include the `tools/stone-pickaxe` goal.

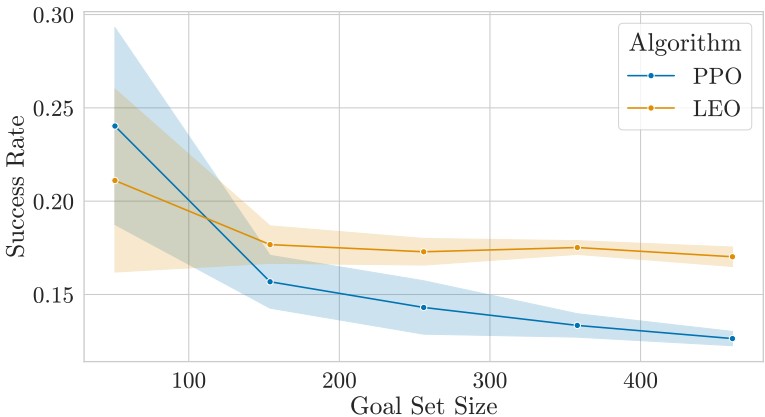

*Figure 18.* Results on CraftaxGC with subsampled goal sets after training for 1 billion timesteps, averaged over all goals. Shaded area denotes 1 standard error over 5 seeds.

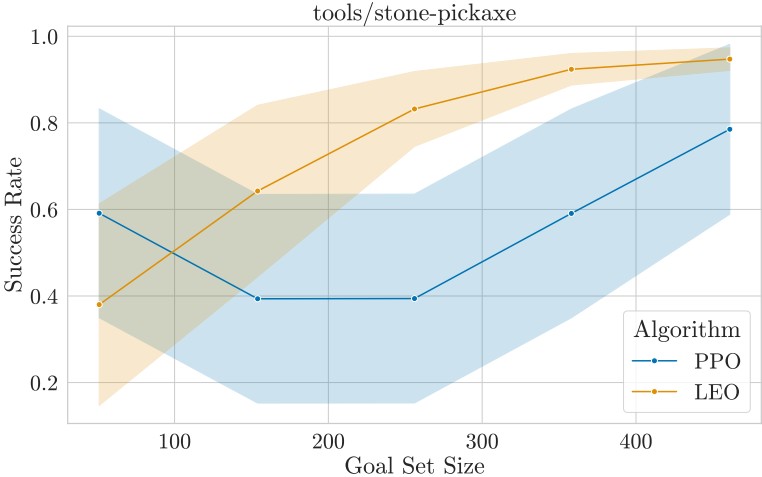

*Figure 19.* Results on CraftaxGC with subsampled goal sets after training for 1 billion timesteps for the `tools/stone-pickaxe` goal. Shaded area denotes 1 standard error over 5 seeds.

# H. Dual Leo Components Further Results

Continuing our analysis of Dual LEO (PQN) from Section 6.2, we investigate the results of acting greedily with respect to both the LEO and UVFA networks when commanding every goal in Craftax-Classic, taking a checkpoint that has trained for 50 million timesteps (Figures 20 and 21).

We see that for most goals with high success rate, the UVFA network has a higher success rate, whereas for lower success rate goals that are still being learned, the LEO network often performs better. For the 9 goals with the lowest non-zero success rate, only the LEO network can successfully solve them. These overall results track with our hypothesis that the LEO network is useful for initial learning, while we can generally expect the UVFA network to converge to a higher success rate.

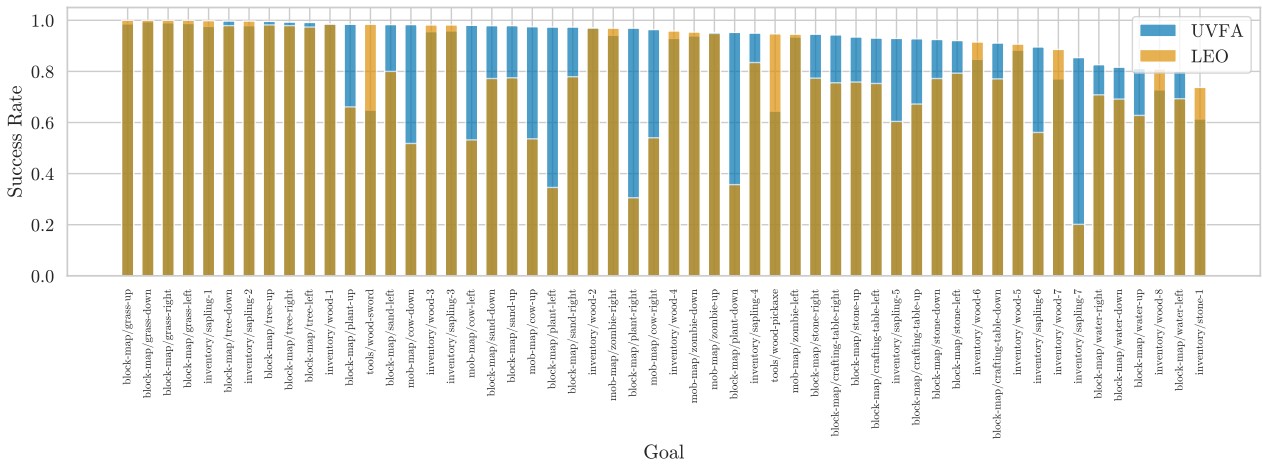

*Figure 20.* Per-goal results for Dual LEO (PQN) when acting greedily with respect to each of its components on Craftax-Classic at 50 million timesteps. Part 1.

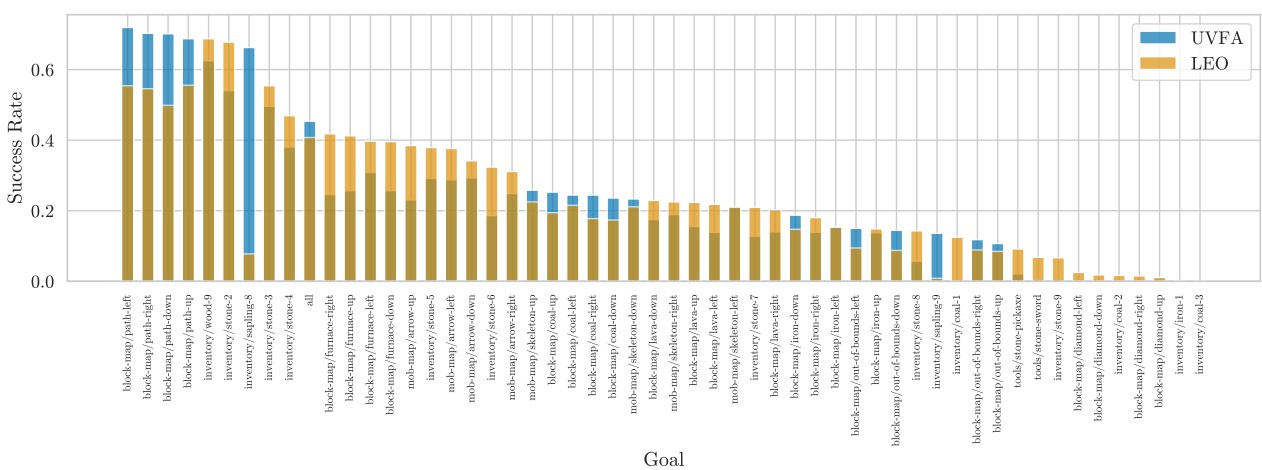

*Figure 21.* Per-goal results for Dual LEO (PQN) when acting greedily with respect to each of its components on Craftax-Classic at 50 million timesteps. Part 2.

## I. Do we need to update with respect to *all* goals?

Updating with respect to all goals may be infeasible in domains where evaluating the reward function for each goal is expensive. For this reason it may be desirable to only perform a partial LEO update, where only a subset of the LEO heads receive signal at every learning step.

To evaluate the feasibility of this, we run experiments in Craftax for 1 billion timesteps where at each update we mask out a random fixed proportion of the per-head Q-value losses (Figure 22). In other words, we perform a many-goals update rather than an all-goals update.

We see that performance smoothly degrades as the proportion and information content of the LEO update decreases. Interestingly, Dual LEO (PQN) seems to saturate at around 60% of heads being updated, indicating this could be a viable strategy.

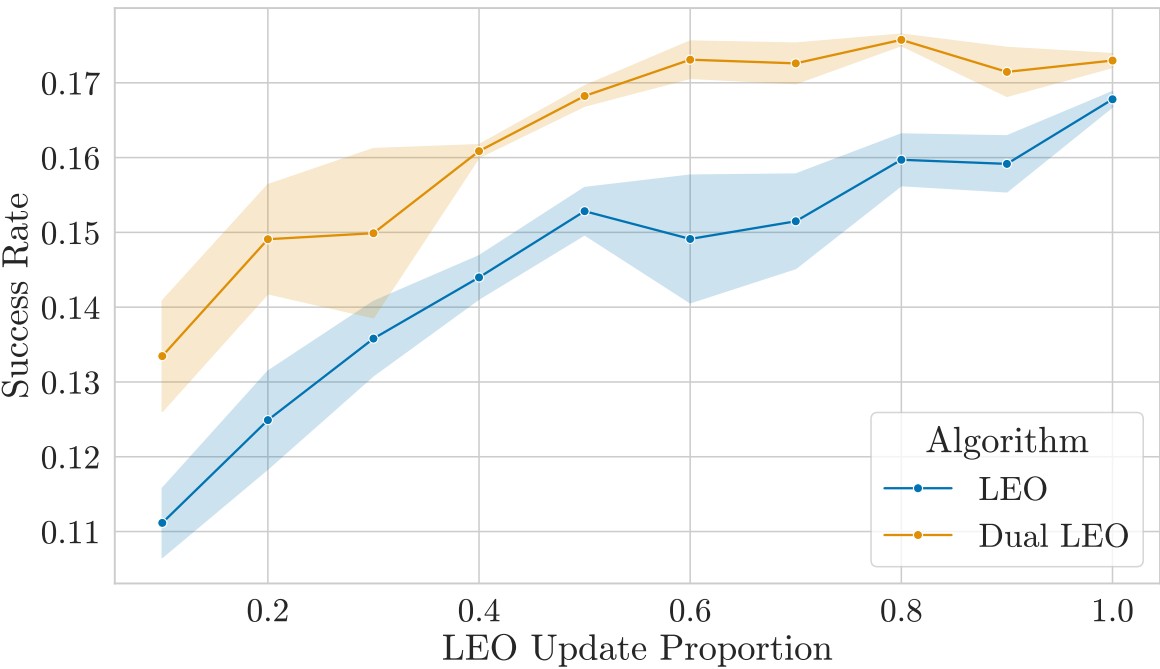

*Figure 22.* Mean goal success rate at 1 billion timesteps on Craftax for LEO and Dual LEO (PQN) when updating with respect to only a random subset of goal heads. For Dual LEO we only modify the LEO update. The shaded area denotes standard error over 4 seeds.

