# OpenReview forum: "Goal-Conditioned Agents that Learn Everything All at Once"
_ICML.cc/2026/Conference — ICML 2026 regular_

### Official Review · Reviewer_MGJr · 2026-02-26

**Soundness:** 3
**Presentation:** 3
**Significance:** 2
**Originality:** 2
**Overall Recommendation:** 4
**Confidence:** 3

**Summary:**

This work introduces Learning Everything all at Once (LEO), a goal-conditioned reinforcement learning (GCRL) framework designed to better leverage rollouts during training.
A central challenge in GCRL is extracting useful signal from trajectories that do not achieve their intended goals.
LEO addresses this by performing all-goals updates, enabling the agent to extract a learning signal for every goal from each experience.
At execution time, an action for a given goal is selected from the action predictions corresponding to all goals.
However, this induces an information bottleneck during training: since the specific goal is only used at the final selection step, the network is implicitly encouraged to learn representations that support all goals simultaneously, rather than specializing to the given one.
To mitigate this, the authors augment the method with a standard goal-conditioned policy and linearly interpolate the Q-value estimates of both networks for action selection.
The paper also introduces a new GCRL benchmark based on Craftax, where goals are framed as high-level tasks rather than target states.
Experiments in both discrete and continuous domains show performance that is comparable to or better than the considered baselines.

**Compliance With Llm Reviewing Policy:**

Affirmed.

**Final Justification:**

In their rebuttal, the authors have provided evidence that (A) their approach scales up to ~10k goals (Figure 1), and (B) decreasing the randomly sampled goal set size leads to worse performance (Figure 2), which better motivates the proposed all-goals update. Based on this new evidence, I increase the Originality score from 1 to 2 since (A) addresses my concern about scalability. Similarly, I also increase the Soundness score from 2 to 3 since (B) offers a principled empirical justification for the proposed all-goals update method. Combined with the clarification regarding the CraftaxGC benchmark, I raise my overall score from 2 to 4.

**Key Questions For Authors:**

1. Have the authors experimented with running LEO using randomly sampled subsets of goals and currying goal-conditioned Q-networks with respect to those subsets? This seems like a more realistic setting with greater potential to scale to large goal spaces. In particular, it would be interesting to understand how performance varies as a function of the size of the sampled goal subset. Is there a specific reason this direction was not explored in the paper?
2. Do the authors agree with my comment regarding the goals defined for Craftax? As I understand it, if goals are verified purely from observations without introducing latent variables, then they still correspond to target sets of states. Am I misunderstanding this point? I would appreciate clarification on whether and how these goals go beyond target-state specifications.
3. Can the authors comment on whether the papers listed above [1, 2] are relevant to their work? In particular, do the authors agree that goals defined via formal specifications (e.g., temporal logic) genuinely go beyond target-state goals, in the sense that they define sets of behaviors or traces rather than sets of terminal states?

**Limitations:**

yes

**Strengths And Weaknesses:**

The paper is generally well written and easy to follow, with a clear motivation and clean exposition of the method.
The introduction of a new GCRL benchmark based on Craftax is a valuable contribution, as standardized and diverse benchmarks are still limited in this area.
The experimental evaluation is reasonably designed to support the main claims, covering both discrete and continuous domains and including appropriate baselines for comparison.


The contribution appears somewhat incremental; conceptually, it is not entirely clear how far the method goes beyond currying Q-functions over goals.
More importantly, as the authors themselves acknowledge, the approach cannot scale well to large goal sets.
Since the end goal of GCRL is, broadly speaking, to generalize to large or even infinite goal spaces, this limitation is significant.
The current formulation curries the Q-function with respect to the entire goal set, which restricts applicability to relatively small or toy domains.
A more scalable alternative, such as currying with respect to random goal subsets or some form of structured goal sampling, would have strengthened the practical impact of the paper.
Also, the benchmark goals are not clearly specified in the appendix; only learning curves are provided.
From the goal names in the training plots, it appears that most goals involve collecting a certain number of items, acquiring tools, or blocking parts of the map.
As the authors state that goals are identified directly from observations without latent variables, these still correspond to target sets of states (e.g., "collect 5 woods" defines states where the wood counter equals 5).
Truly high-level tasks that go beyond target states, for example, could use latent variables and temporal abstractions specifying behaviors, i.e., traces, rather than terminal state sets.
As presented, the benchmark goals do not seem to fully achieve this distinction.
Additionally, there is a body of related work that is not discussed.
In particular, similar problems have been studied in specification-guided RL, where goals are represented as formal specifications such as temporal logic formulas or automata, defining a set of behaviors for each goal rather than target states.
I believe these works should be discussed at least in the related work section.
Please see below for a non-exhaustive list of such papers.

1] Vaezipoor, Pashootan, et al. "Ltl2action: Generalizing ltl instructions for multi-task rl." ICML 2021.

[2] Yalcinkaya, Beyazit, et al. "Compositional automata embeddings for goal-conditioned reinforcement learning." NeurIPS 2024.

---

> ### Author Rebuttal · Authors · 2026-03-31
>
> Thank you for taking the time to read and review our submission and for raising some informed points, which we now address:
>
> # Significance of LEO
> The reviewer stated that it is “not entirely clear how far the method goes beyond currying Q-functions”. We would argue that the currying of the Q network (and policy network in the actor-critic case) is a reformulation that allows for efficient all-goals updates, rather than an end to itself. Updating with all goals is largely underexplored and goal relabelling is almost always viewed through the limited perspective of HER. The chief insight of our work is into the inherent tradeoff between LEO (efficient all-goals updates, limited fidelity due to late fusion) and UVFA (inefficient updates, high fidelity) and how best to combine these two approaches (i.e. Dual LEO).
>
> Below, we also show that Dual LEO scales to ~10k goals (a regime where relabelling-based methods are completely impractical) and that LEO and Dual LEO degrade gracefully when partially curried.
>
> # Goal set size
> The reviewer is right in pointing out that LEO is indeed limited to finite goal sets that are small enough to reasonably curry. While we acknowledge this limitation, the practical scope of LEO goes far beyond “small or toy” domains. Craftax is certainly not a toy environment, remaining an unsolved benchmark to this day. Furthermore, the size of the goal set is not necessarily an indication of difficulty, otherwise the implication is that single-task RL is trivial.
>
> As a demonstration, we run experiments on a much larger goal set in Craftax, where we take every possible setting of every 1-hot vector and set it as a goal, giving a goal set size of 9637 ([Figure 1](https://anonymous.4open.science/api/repo/LEORebuttal-25BB/file/all_tokens.png)). We see on this much larger goal set that the late fusion issue discussed in Section 3.2 becomes even more pronounced and LEO alone struggles as it must spread its computation over ~10k goals. However, Dual LEO outperforms PPO, validating that the ideas behind Dual LEO scale to a goal set an order of magnitude larger than our main experiments.
>
> We have added this experiment to the manuscript and believe it strengthens the paper.
>
> # Partial goal set currying (Q1)
> The idea of currying with respect to subsets of goals is interesting, and we agree would make LEO more feasible for even larger goal sets. We run experiments for both LEO and Dual LEO where each transition is used only to update a random subset of the goals.
>
> [Figure 2](https://anonymous.4open.science/api/repo/LEORebuttal-25BB/file/many_goal_updates.png) shows final performance over the whole goal set on Craftax after training for 1 billion timesteps as we vary the proportion of the LEO network that we curry and update with respect to on each update. The results show that both LEO and Dual LEOs performance degrades gracefully as we update with respect to fewer goals, indicating that this approach would be effective for learning on larger goal sets. We have also added this result to the manuscript.
>
> # Goal set listing
> Thank you for pointing out that the Craftax goal set is not actually listed in the appendix (aside from implicitly in the plots) - we have updated the manuscript with a full table of goals.
>
> # CraftaxGC as a state-based benchmark (Q2)
> Finally, the reviewer is correct that CraftaxGC goals all correspond to a set of states, however the subtle yet crucial distinction is that different goals are inferred from different parts of the observation, meaning that many goals can be fulfilled at the same time, breaking the functional mapping assumption that is common in GCRL (e.g. this is assumed in both HER and Contrastive RL).
>
> # Goals as formal specifications (Q3)
> We agree with the reviewer that LTL2action [1] and cDFAs as goals [2], as well as the broader field of representing goals with temporal logic, do go beyond simple functional target state goals. Our claim was not meant to be that CraftaxGC is entirely unique in this aspect, but rather that this is a simplification that the vast majority of work in GCRL falls into. The reviewer is correct that our motivation for CraftaxGC in Section 4 stated this too strongly and we have added the "most" qualifier to our claim. We have also added a section to the related work discussing goals specified with temporal logic, as well as natural language, as other examples of non-functional goal settings.
>
> # Conclusion
> We thank you for your insightful suggestions, and believe that the manuscript will now be stronger as a result. If you have any further questions/suggestions we will happily engage. Otherwise, if our response has satisfied you, we ask you to consider raising your score.
>
> Best wishes,
>
> The authors
>
> [1] Vaezipoor, Pashootan, et al. "Ltl2action: Generalizing ltl instructions for multi-task rl." 2021.
>
> [2] Yalcinkaya, Beyazit, et al. "Compositional automata embeddings for goal-conditioned reinforcement learning." 2024.

---

> > ### Author Rebuttal · Reviewer_MGJr · 2026-04-01
> >
> > I thank the authors for their response and for addressing my comments. I will increase my score accordingly.

---

### Official Review · Reviewer_jTcT · 2026-03-13

**Soundness:** 3
**Presentation:** 4
**Significance:** 3
**Originality:** 4
**Overall Recommendation:** 5
**Confidence:** 4

**Summary:**

The authors introduce an approach to goal-conditioned RL called that allows agents to learn values and policies for every goal at once given any state transition. In contrast to prior approaches like UVFA, which maps states and goals to value functions, they take the approach of mapping states to value functions for a set of goals. This change allows for a vectorised Bellman update that can update the value functions for all goals using any (s,a,s') transition data -- but contingent upon having reward labels for all goals -- allowing for more efficient use of the training data. They validate their approach with a series of experiments in procedurally generated domains as well as in AntMaze tasks and show that a dual approach that linearly interpolates between LEO and UVFA performs best overall.

**Compliance With Llm Reviewing Policy:**

Affirmed.

**Key Questions For Authors:**

I would be interested in seeing how performance on a given goal changes for each algorithm as the total number of goals learnt increases. Do we see signs of transfer or interference between goals for each algorithm? Does it depend on the algorithm, the number of goals and/or the model capacity?

**Limitations:**

yes

**Strengths And Weaknesses:**

**Strengths**:

The proposed approach is elegant and technically sound, and the presentation is very clear. The approach provides a clear contribution as it complements prior approaches to goal-conditioned RL, particularly UVFA and HER. I also appreciated that the authors' discussion of the nuances of their approach including, for instance, information bottleneck issues as well as the teacher-student dynamics between LEO and UVFA in their dual LEO variant. And the authors have provided careful experimental validation of their approach, comparing against appropriate baselines including UVFA and HER algorithms. Their approach naturally decouples the behaviour policy from the target policy for each of the goals, something that presents the opportunity to explore the best behaviour policy for multi-goal learning. Overall, I am inclined to support acceptance of the paper.

**Weaknesses**:

The weaknesses listed. below have more to do with algorithmic limitations than anything else, some of which the authors have already discussed in their papers. In my view, there is no critical weakness that would prevent this paper from being accepted.

One constraint of LEO is the need to have access to reward labels for all goals, not just the commanded goal, but this may in fact be the case in standard multi-task RL settings.

LEO's approach of representing goals as indices in an output tensor poses certain limitations. As the authors themselves pointed out, this can lead to information bottleneck issues as the model must represent information for all goals until the final layer when the commanded goal becomes evident. And when goals are continuous, the space of goals must be discretised. But in comparison to UVFA, there seems to be another limitation to LEO, and that is that their setup seems to limit the potential for generalisation across similar goals. The space of goals is parametric, as in continuous navigation tasks like antmaze, UVFA's approach of representing value functions as a function of goals more naturally allows for the potential to generalise over nearby goals and to allow transfer of knowledge from one goal to another. This benefit seems lost in LEO's vectorised approach which cannot model any relational information between goals.

Relatedly, I would be interested in seeing how learning of other goals affects performance on a given target goal. For instance, in Fig 4, increasing the number of goals from 136 to 512 does make LEO into the best overall algorithm, but the average success rate drops significantly. For a given goal in the set of classic goals, I would be interested in seeing how its performance changes for each algorithm when moving from 136 to 512 goals. Does the expanded goal set improve performance on the target goal because of transfer, do we see interference instead, or is it algorithm dependent?

Another limitation of LEO is that it requires a careful curation of the goal set to represent. For instance, if a goal has very sparse rewards, the corresponding Bellman update will remain zero until the agent encounters a transition with non-zero reward for that goal. Thus, without a careful curation of the goal set, many elements in the vectorised Bellman update could end up being zero for most of the time. The authors address this somewhat with a goal filtering technique, where commanded goals are selected only from previously experienced goals, as well as the "first return then explore" exploration strategy for handling sparse rewards.

---

> ### Author Rebuttal · Authors · 2026-03-30
>
> Thank you for reviewing our work. We appreciate your view that our submission is a “clear contribution” and for you advocating acceptance of the paper. We now address your questions:
>
> # Oracle access to reward function
> We do implicitly assume oracle access to the goal-conditioned reward function, but this is common across essentially all methods that do some form of goal relabelling (e.g. HER).
>
> # Representing goals as indices
> The reviewer is correct that LEO does cause a loss of inductive bias due to the goals being represented simply via indexing. This is not an issue in settings like Craftax with heterogeneous goal sets, but this is a valid point for environments like the ant mazes where the true goal set does have some ignored underlying structure.
>
> Firstly, ignoring the structure can actually be a good thing. Two positional goals in the ant maze might be close together in 2D space, but exist on opposite sides of a wall and need completely different paths to reach them. In this case, UVFA style approaches have to learn sharp discontinuities in the goal space (with some potential interference), while each LEO head can simply learn on its respective goal. The actual structure of the goal set is induced by the MDP dynamics, not the geometry of the goal space, so in some cases UVFA-style approaches can actually be deceiving.
>
> Secondly, while the learning signal for each head is decoupled, they will all contribute to shared representation learning of the earlier layers.
>
> Finally, each head is still learning on every transition, so in theory it shouldn’t actually need the inductive prior of some structured goal space. From the perspective of each head, it has seen and updated itself with every transition so far encountered.
>
> All that said, this is a valid point and we will add the following sentence to the limitations section: “Also, LEO represents goals as a set, which may be suboptimal if the goals have some underlying structure, for instance representing points on a grid.”
>
> # How does the goal set size affect learning for an individual goal?
> This is a very interesting question. To answer this we took the full Craftax goal set (512 goals) and ran experiments with random subsets of various sizes for both LEO and PPO.
>
> Aggregating over all goals, we see that the general trend is that larger goal sets are harder ([Figure 1](https://anonymous.4open.science/api/repo/LEORebuttal-25BB/file/partial_goal_set_all_goals.png?v=8f3ed6c7)). However, we see that PPO is far more affected by this than LEO, which makes sense as all-goals learning will become more of a differentiator as the goal set size increases.
>
> However, looking specifically at the stone_pickaxe goal, we actually see that performance increases as the goal set gets larger ([Figure 2](https://anonymous.4open.science/api/repo/LEORebuttal-25BB/file/partial_goal_set_stone_pickaxe.png?v=bc7060b8)). In our setup, both exploration and learning are intrinsically tied together. A smaller goal set is potentially easier to learn on with respect to a fixed dataset, but may lack ‘stepping stone’ goals that encourage exploration. For medium difficulty goals like stone_pickaxe, it seems that the benefits of stepping stones outweigh the need to learn on a larger goal set.
>
> # Goal set curation
> We would actually argue that LEO is more resilient to uncurated goal sets than other methods, as shown in Figure 16 where the gap between LEO/Dual-LEO and the baselines is even larger for Craftax when the goals are not curated through a simple curriculum.
>
> Most updates being zero is not necessarily a bad thing - the framing of HER can lead one into thinking that learning is only done on ‘positive’ samples, but negative samples are still absolutely necessary to learn on, although in many cases their information content might be lower. For instance, imagine doing a ‘maximal HER’ algorithm where every transition is relabelled with the positive goal at that timestep - the Q function would learn an optimistic bias as it would never see negative examples.
>
> # Conclusion
> Thank you for your questions. We think that the modifications to the manuscript, especially the experiment about varying the goal set size, has improved the framing of our paper. We would be happy to engage with any further questions you might have.

---

> > ### Author Rebuttal · Reviewer_jTcT · 2026-04-04
> >
> > I thank the authors for their detailed rebuttal. I'll address each of the authors' responses below.
> >
> > **Oracle access to reward function:**
> > I thank the authors for their clarification here.
> >
> > **Representing goals as indices:**
> > I fully agree with the authors' maze wall example, though I would argue that the issue is in part a problem of finding an appropriate parametrisation of the goal space. Consider the example of a hairpin maze (see e.g. https://maze.conductscience.com/wp-content/uploads/2017/06/Hairpin_maze_02.jpg) where the "maze" consists of just a single path that winds back and forth on a 2D surface, going from one corner to the opposite. One could parametrise this using 2D Euclidean coordinates, in which case two points on opposite sides of a wall would appear close together. Alternatively, one could use a 1D coordinate system that simply marks how far along the path a point is. In this case, points that are close together in this coordinate system would also be close together in terms of the shortest path connecting them while points on opposite sides of a wall would be distant in this parametrisation. So while the structure of the goal set is determined by the transition dynamics, the issues the authors point out stem in part from an inappropriate parametrisation of the goal/state space. While addressing this problem is beyond the scope of the authors' work, I'm not entirely convinced it'll be beneficial to ignore the topological structure of the goal space when such structure exists.
> >
> > **How does the goal set size affect learning for an individual goal:**
> > I thank the authors the additional experiments. The results and the authors' additional comments are illuminating, and I have no further questions or comments here.
> >
> > **Goal set curation:**
> > I guess this depends on how the Q-values are initialised before learning starts. Learning is driven by prediction errors, and if the Q-values started off at zero, the prediction errors will be zero until the agent encounters the corresponding goal. One should not expect the Q-values to update before that happens, otherwise, it should also update in the hypothetical situation where there actually is no goal. On the other hand, if the Q-values began overly optimistically or pessimistically, the negative transitions will update the Q-values toward zero.
> >
> > That being said, my point remains. The agent cannot conceivably learn from a goal, even indirectly via bootstrapping, until it first encounters the goal. And this could be an issue should the goal set include many difficult-to-encounter goals.

---

> > > ### Author Response · Authors · 2026-04-04
> > >
> > > Thank you for your response and for raising some valid questions, which we now address:
> > >
> > > # Representing goals as indices
> > >
> > > This is a fair point. A goal representation space can provide a useful inductive prior if it maps onto the MDP dynamics, but can be misleading if it does not. The hard part in many cases is finding a good structure (there is some interesting recent work related to this [1]). LEO in its basic form essentially sidesteps this entire issue and makes no assumptions about the underlying dynamics. Our claim was certainly not meant to be that ignoring this structure is always (or even often) a good thing and we will reflect this in the additional sentence to the limitations section mentioned in our initial rebuttal.
> > >
> > > # Goal set curation
> > >
> > > This is also true. In the case of binary terminal goal sets we could initialise all goal outputs to zero, and then be certain that no learning signal will pass through any head until its respective goal has been observed. In practice, we do find that LEO seems to be quite robust to this (see [Figure 1](https://anonymous.4open.science/api/repo/LEORebuttal-25BB/file/all_tokens.png) in our rebuttal for reviewer MGJr where we train on a ~10k goal set, in which only ~3k goals are ever observed, meaning that 2/3rds of all LEO updates could theoretically be ignored). Intuitively, as the Q-estimates approach zero, the bootstrapping values likely also approach zero, meaning that the TD error for these heads would be very small and contribute very little to the overall loss. That being said, we will note that many heads will never see a positive signal explicitly as a limitation in the paper.
> > >
> > > Thank you again for engaging with us in our submission, and we hope you found our conversation interesting!
> > >
> > > [1] - https://arxiv.org/abs/2510.06714

---

### Official Review · Reviewer_UHC1 · 2026-03-13

**Soundness:** 3
**Presentation:** 3
**Significance:** 2
**Originality:** 3
**Overall Recommendation:** 3
**Confidence:** 3

**Summary:**

This paper proposes Learning Everything all at Once (LEO), a goal-conditioned RL method that reuses each collected transition to learn all possible goals. Instead of training on augmented transitions with relabeled goals, LEO predicts Q values for every goal in a single forward pass, which enables an all-goals update. The approach achieves strong performance while providing over 250x speed-up compared to baselines.

**Compliance With Llm Reviewing Policy:**

Affirmed.

**Final Justification:**

The authors have addressed weakness 1, 2, 4, and 5 in their rebuttal. Weakness 4 is not fully addressed, so I will keep maintain by current score.

I appreciate the additional experiments on the greedy evaluation of LEO vs. UVFA, though the results in this experiment seem to show that UVFA outperforms LEO, contradicting the core results in the main paper. I like the idea of LEO, though a research contribution requires more than just showing a new method outperforms existing baselines: it requires evidence that the gains stem from the proposed mechanism. In other words, we must show the algorithm does what it claims to do, and as a result, improves performance.

The paper would made a good contribution with additional experiments that highlight precisely *why* LEO works.

**Key Questions For Authors:**

1. Is there a less heuristic way to ensure LEO does not under optimize easy goals?
1. Why specifically does LEO under-perform in Figure 1? It's apparent that the setup is simple enough for standard baselines, but I would then assume that it should also be simple for LEO?
1. Can the authors provide additional experiments showing *why* LEO improves performance as well as a cost comparison?

**Limitations:**

Limitations are discussed throughout the text.

**Strengths And Weaknesses:**

**Strengths**
1. LEO is conceptually intuitive and well-motivated. The algorithm description is easy to follow.
2.  Empirical evaluation is clean. The chosen environments highlights various strengths and weakness of LEO as well as baseline methods and are complex enough that there is a sizable performance difference between methods.

**Weaknesses**
1. Figure 14 shows that PPO and PQN + HER outperform LEO, and PPO outperforms Dual LEO. While the authors explain that why all-goal learning may not be necessary to achieve strong performance in simpler tasks, it makes me wonder why LEO does not perform well in these supposedly simpler settings.
1. Linearly interpolating the Q-value estimates of the LEO network and a goal-condition network seems to work empirically but is a heuristic without much motivation.
1. The extension of LEO to continuous settings is a discretization approximation. As written LRO does not natively handle continuous goals.
1. The empirical evaluation does not directly assess the quality of the Q-values learned by LEO compared to a standard goal-conditioned network. A central motivation of LEO is that it enables more efficient learning of accurate Q-values, yet this claim is not explicitly validated in the experiments. More broadly, while the results demonstrate improved performance in most settings, it would be valuable to include analyses that clarify why LEO improves performance and whether the gains arise from the mechanisms described by the authors when motivating LEO.
1. I was hoping to see some discussion and experiments on the computation cost of LEO vs a goal-conditioned network, though this is not included.

---

> ### Author Rebuttal · Authors · 2026-03-30
>
> Thank you for taking the time to read our submission. We address your raised points as follows:
>
> # Why does LEO underperform in some cases? (W1 + Q2)
> There is an inherent tradeoff between policy fidelity and the number of goals updated: UVFA has high fidelity but updates one goal at a time, while LEO updates all goals at lower fidelity due to late fusion. In simpler environments the benefits of all-goals learning are less relevant, and therefore the late fusion issue becomes more pronounced. As for why Dual LEO is beaten by PPO in this setting, Dual LEO is based off of PQN, which is generally a weaker algorithm than PPO.
>
> # Justification for Dual LEO (W2)
>
> We investigate how the individual networks of Dual LEO perform when taken by themselves and evaluated over goals. That is to say that we train the two networks on the same stream of data (acting with a linear combination over Q values) as normal with Dual LEO, but then periodically test each network by itself. We show the results on two goals ([Figure 1](https://anonymous.4open.science/api/repo/LEORebuttal-25BB/file/dual_leo_components_coal.png) and [Figure 2](https://anonymous.4open.science/api/repo/LEORebuttal-25BB/file/dual_leo_components_stone.png)).
>
> These results shed some light onto the subtleties of LEO. Firstly, we see that on these (medium difficulty) goals, LEO starts succeeding at them before the UVFA network. This validates our claim that LEO acts like a “data sponge”, by internalising how every observed transition relates to each goal. However, we also see that LEO converges to a lower success rate, due to the late fusion issue that we discuss in the manuscript. In contrast, the UVFA network only starts making progress on these goals once the LEO network has induced enough positive examples to train on, at which point the UVFA network overtakes LEO and converges to a much higher success rate due to its superior representational capacity.
>
> Dual LEO acts just as we proposed it would in Section 3.2, effectively taking the max of its two constituent components.
>
> One key point to note is that PQN by itself never learns to solve either of these goals - validating our claim that LEO acts as a ‘teacher’.
>
> # Q-Value Quality (W4/Q3)
>
> To directly investigate Q-value quality, we record the MSE between each Q-function and the Monte Carlo returns of its own greedy policy, for both the LEO and UVFA networks in Dual LEO ([Figure 3](https://anonymous.4open.science/api/repo/LEORebuttal-25BB/file/q_val_acc.png)). Note that this measures self-consistency (how well each network predicts the returns of its own policy) rather than proximity to the optimal Q*, which is generally unknowable. The UVFA network achieves lower error, consistent with our discussion of late fusion in LEO. However, as shown in the previous section, LEO's Q-values are still 'directionally' accurate enough to achieve goals before UVFA can, making LEO effective as a teacher even if its value estimates are less precise.
>
> # Alternatives to Dual LEO (Q1)
>
> Other approaches to using LEO as a teacher could be to keep track of success rates for each goal and simply act with the best network, or instead use this to inform a per-goal mixing value. Alternatively, you could use LEO for behaviour cloning e.g. combined with PPO. These would all be exciting future directions to explore.
>
> # Discretisation Approximation (W3)
>
> LEO cannot natively handle continuous goal spaces, but we have shown that it can actually achieve reasonable performance through the discretisation approximation, outperforming UVFA style approaches. In these settings the tradeoff with LEO is not just about all-goals learning vs network fidelity, but also with respect to the accuracy of the goal space approximation, where the density of goals (e.g. grid density in the ant mazes) can be tuned appropriately. Furthermore, as noted in Section 5.2, LEO could be extended by adding the residual error to the input.
>
> # Computational Cost (W5 + Q3)
> We discuss computational cost in Figure 2 in the manuscript, where we show that LEO is significantly more computationally efficient than all-goals learning via relabelling, and is only marginally slower than learning with respect to a single goal. Running for 1 billion timesteps on a single L40S GPU, PQN takes 5 hours, LEO takes 7 hours and PQN+HER takes 11 hours. LEO is only ~40% slower to learn with all 512 goals, while PQN+HER is twice as slow to only relabel with 2 extra goals. Despite this, LEO massively outperforms PQN+HER on Craftax.
>
> # Conclusion
> We thank the reviewer for their questions and think that the deep dive into Dual LEO has especially strengthened the manuscript, as it empirically validates our motivation for Dual LEO which we had previously only hypothesised. If the reviewer has any further questions/suggestions we would be happy to respond. Otherwise, if you judge our rebuttal to have satisfied you, we ask you to consider raising your score.
>
> Best wishes,
>
> The authors

---

> > ### Author Rebuttal · Reviewer_UHC1 · 2026-04-03
> >
> > Thank you for your response. I'm satisfied with everything except the additional Q-value experiments. If the claim is that LEO is improving the "directional accuracy" of the Q values, then this claim should be supported empirically. To clarify, it's fine that the MSE does not decrease toward zero for LEO, as we can still derive an optimal policy even if Q-value MSE is large (e.g. $argmax_{a\in A} [Q*(s, a) + c]$ is an optimal policy for any constant $c$).
> >
> > One way to quantify directional accuracy might be: sample arbitrary states $s$ and then compute the fraction of states in which the argmax of the LEO / UVFA networks is the optimal action. This metric unfortunately requires access to $Q*$, but perhaps this experiment could be run in a toy environment where $a^*$ is easy to compute.  A proxy metric could be: compare the returns of the greedy policies induced by the LEO / UVFA networks. The authors may see other approaches. In any case, I'd like to see support that LEO's performance improvements are due to better Q estimation.

---

> > > ### Author Response · Authors · 2026-04-04
> > >
> > > Thank you for your response and for so thoroughly engaging with our submission. To go into more detail on the Q-value accuracy point we provide extra experiments in a simple Gridworld maze where we have access to $\pi^* $, as well as investigating how the greedy LEO and UVFA policies behave on Craftax.
> > >
> > > # Gridworld Maze
> > > We create a simple goal-conditioned gridworld maze environment where each non-wall location is a goal and evaluate over three different map sizes ([map_size_12](https://anonymous.4open.science/api/repo/LEORebuttal-25BB/file/grid_12.png), [map_size_16](https://anonymous.4open.science/api/repo/LEORebuttal-25BB/file/grid_16.png) and [map_size_24](https://anonymous.4open.science/api/repo/LEORebuttal-25BB/file/grid_24.png)). The player can move in the 4 cardinal directions and cannot move through walls. The player always starts in the middle of the maze. Note that not all goals are reachable. We precompute the optimal policy $\pi^*$ by running a BFS from every starting point.
> > >
> > > We run Dual LEO and measure the proportion of actions for which $\pi^*(s,g) = \text{argmax} Q(s,g)$ for both LEO and UVFA networks throughout training over 5 seeds. We use $\gamma<1$ (0.99) so our greedy policies should converge to the shortest path.
> > >
> > > We see that, for all maze sizes, the LEO component of Dual LEO has a higher proportion of greedy actions matching the optimal actions throughout training ([Figure 4](https://anonymous.4open.science/api/repo/LEORebuttal-25BB/file/optimal_policy_match.png)). For completion, we also show the overall success rates on each map size ([Figure 5](https://anonymous.4open.science/api/repo/LEORebuttal-25BB/file/gridworld_success_rate.png)).
> > >
> > > # Craftax
> > > In Craftax, we do not have access to either $\pi^* $ or $Q^*$ but, as you pointed out, we can use proxy values to provide some insight. We look at the returns of the greedy policies induced by both LEO and UVFA networks and consider their success rate for every goal at 50M timesteps on Craftax-Classic over 5 seeds ([Figure 6](https://anonymous.4open.science/api/repo/LEORebuttal-25BB/file/all_goals_craftax.png)). We see that, while the greedy UVFA network is better overall than the greedy LEO network, there are many goals that LEO is better on, including some that the UVFA network completely fails to solve. We believe that this, alongside Dual LEO significantly outperforming both LEO and UVFA by themselves is solid evidence for our claim.
> > >
> > > Thank you for your questions and we hope that our response and additional experiments have been satisfactory. We will incorporate these results into the manuscript.

---

### Decision · Program_Chairs · 2026-04-30

**Decision:**

Accept (regular)

**Comment:**

The paper introduces LEO, a goal-conditioned RL framework that performs an all-goals Bellman update on every environment transition. Rather than conditioning a Q-network on the goal via a state-goal embedding (UVFA), LEO curries (in the functional sense) the Q-function over a finite goal set and predicts Q-values for every goal in a single forward pass, enabling the simultaneous update of all goal-specific Q values from a single (s,a,s’) transition. The paper also introduces Dual LEO, which linearly interpolates LEO’s Q-estimates with those of a standard UVFA network, and contributes CraftaxGC, a new goal-conditioned benchmark built on Craftax.

Reviewers recognized that LEO is elegantly formulated, that it complements rather than replaces UVFA and HER, and that the authors’ own discussion of the information bottleneck and teacher-student dynamics is unusually honest. The rebuttal substantially strengthened the paper’s empirical foundations in three ways.

First, on scalability. MGJr’s central concern was that LEO cannot scale to large goal sets. The authors responded with a new ~10k-goal Craftax experiment (every 1-hot vector setting as a goal), showing that (i) LEO alone struggles at this scale as the late-fusion bottleneck worsens, but (ii) Dual LEO still outperforms PPO. They also added a partial-currying sweep where each transition updates only a random subset of heads; both LEO and Dual LEO degrade gracefully, suggesting partial currying as a practical path to even larger goal sets. Based on this evidence, MGJr raised the overall recommendation from 2 to 4.

Second, on the teacher-student interpretation of Dual LEO. UHC1 asked why the linear interpolation works, beyond empirical observation. The authors responded with a component analysis: they train Dual LEO normally but periodically evaluate each constituent network in isolation (Figures 1-2 of the rebuttal). The result is: LEO acts as a “data sponge” and succeeds on medium-difficulty goals (coal, stone) before the UVFA network learns anything, then the UVFA network overtakes LEO and converges to a higher success rate, and Dual LEO effectively takes the max of the two. This empirically validates the teacher-student intuition that was previously only hypothesized.

Third, on Q-value quality. UHC1 originally noted that the paper claims better Q-learning but never measures Q-value accuracy. The authors responded first with a self-consistency experiment (MSE against Monte Carlo returns of each network’s own greedy policy) which UHC1 correctly noted is a weaker claim than directional accuracy. The authors followed up with a gridworld experiment where optimal policies can be computed by BFS, and show that LEO’s greedy action matches the optimal action more often than UVFA’s across three maze sizes throughout training. This is strong evidence.

Reviewer MGJr’s point about CraftaxGC goals being effectively target-state sets (since they are inferred from observation components) is valid but addressed appropriately: the authors accept the critique, soften the claim to “most GCRL”, and add a related-work discussion of specification-guided RL (LTL2Action, compositional automata embeddings) which is sufficient scoping.

The contribution is clearly technically sound, the writing is clear, the empirical evaluation is thorough, and the honest discussion of limitations (information bottleneck, discretization of continuous goals, goal-set curation) makes the paper credible. The rebuttal additions – 10k-goal scaling, partial-currying degradation, Dual LEO component analysis, gridworld optimal-policy-match experiment are all strengthen the paper. I recommend acceptance. I encourage the authors to (i) resolve UHC1’s final Craftax reconciliation question before the discussion closes, (ii) incorporate the 10k-goal experiment and the Dual LEO component analysis into the main paper or a visible appendix, (iii) add the promised specification-guided RL related work, and (iv) add the goal-set table to the appendix.